# Enhancing feelings of security: How institutional trust promotes interpersonal trust

Giuliana Spadaro[1]*, Katharina Gangl[2,3], Jan-Willem Van Prooijen[1,4], Paul A. M. Van Lange[1], Cristina O. Mosso[5]

**1** Department of Experimental and Applied Psychology, Vrije Universiteit Amsterdam, Amsterdam, Netherlands, **2** Department of Economic and Social Psychology, University of Goettingen, Goettingen, Germany, **3** Institute for Advanced Studies (IHS), Competence Centre, Insight Austria, Vienna, Austria, **4** The Netherlands Institute for the Study of Crime and Law Enforcement (NSCR), Amsterdam, Netherlands, **5** Department of Psychology, University of Turin, Turin, Italy

\* g.spadaro@vu.nl

**Data Availability Statement:** All relevant data are within the manuscript and its Supporting Information files.

## Abstract

Interpersonal trust is an important source of social and economic development. Over decades, researchers debated the question whether and how public institutions influence interpersonal trust, making this relationship a much-discussed issue for scientific debate. However, experimental and behavioral data and insights on this relationship and the underlying psychological processes are rare and often inconsistent. The present set of studies tests a model which proposes that institutional trust indirectly affects trust among unrelated strangers by enhancing individuals' feelings of security. Study 1 (survey on trust in a broad spectrum of state institutions), Study 2 (nationally representative data from 16 countries), and Study 3 (experimental manipulation of institutional trust) provide convergent support for this hypothesis. Also, the results show that the effect remains consistent even after controlling for individual dispositions linked to interpersonal and institutional trust (Study 1 and 3) and country level indicators of institutional performance (Study 2). Taken together, these findings inform and contribute to the debate about the relationship between institutions and interpersonal trust by showing that when institutions are trusted, they increase feelings of security, and therefore promote interpersonal trust among strangers.

## Introduction

Trust among citizens is crucial for the societal, political, and economic functioning of a state [1]. Societies with high interpersonal trust have happier citizens [2], more political participation [3], and stronger economic growth [4]. Traditionally, trust is deeply rooted in social interaction in dyads and small groups. However, those interactions do not occur in a social vacuum, as they are directly or indirectly embedded in a societal context regulated by institutions, which constitute "rules of the game" that structure social life [5].

In the last two decades, research has been conducted to understand whether certain features of the institutional setting (e.g., performance, efficiency, and fairness) can favor the

**Funding:** The research was supported by the Austrian Science Fund (FWF; project 24863-G1) awarded to KG. The funding source contributed to proofreading of an earlier version of the manuscript.

**Competing interests:** The authors have declared that no competing interests exist.

development of interpersonal trust among strangers [6–10], often leading to opposite predictions [11]. Also, little empirical evidence for the key mechanisms underlying a presumed association between institutions and interpersonal trust is provided in the research literature (e.g., [12]), largely relying on survey studies within one particular society and a limited set of institutions. In the present research, we contribute to this body of literature by focusing on individuals' perceptions of institutions and, more specifically, testing the hypothesis that institutional trust can stimulate trust among strangers by enhancing feelings of security. Across three studies (a survey, an analysis of a cross-national database, and an experimental study), we tested a model proposing that institutional trust fosters interpersonal trust as it serves as cue conveying that one is not completely at the mercy of potentially hostile strangers, but rather is protected in case strangers have malevolent intentions.

## Interpersonal trust and institutions: Crowding-out and institution-centered approaches

Interpersonal trust is a pervasive phenomenon of social life and a lubricant for many societal processes. It is defined as a psychological state that involves the intention to accept vulnerability in social interactions, under conditions of social risk and interdependence [13,14]. It is often conceptualized as a multidimensional construct that can be differentiated into trusting beliefs and the resulting trusting intentions or behavior [15–17]. The former refer to individuals' perceptions of trustworthiness of others, while the latter reflect the acceptance of vulnerability and actions undertaken to gain possible advantages [18]. Positive expectations regarding the intentions or behavior of one or more persons are often based on own direct experience or reputational information shared by others [19–21]. Such interaction processes are key to the development of trust and, thus, to people's willingness to accept risk in situations often characterized by strong interdependence (e.g., [18,22]; for reviews see [23,24]). However, in modern complex societies, individuals often face interactions with strangers, in situations where trust is unlikely to be based on social interactions or reputational information shared by others. Here formal institutions, conceptualized as external systems of control that offer "rules of the game", help to structure and increase the predictability of such exchanges [5]. Indeed, state institutions such as public administration or the police provide safeguards ensuring that others behave cooperatively, and therefore offer cues that increase trust in others [11].

The relationship between institutions and interpersonal trust has been debated, however, especially in light of institutions' key role in providing rules and normative expectations [7,11]. The main claim of the so-called crowding-out approaches is that interpersonal trust becomes no longer relevant if institutions are in place. External sources of control as sanctioning systems directly affect the incentive structure of the interaction by increasing the cost of non-cooperative actions. As a consequence, such institutions remove the social uncertainty involved in the social exchange, and expectations and actions become driven by assurance rather than interpersonal trust [25]. While interpersonal trust is based on beliefs about the benevolent and intrinsic motivation of the partner, assurance is based on the expectation that the interaction partner will behave according to the relevant incentive structure, thus crowding-out interpersonal trust and voluntary cooperation [26]. Moreover, the mere need to establish such systems can even be considered as a signal of others' untrustworthiness and, therefore, decrease the motivation to trust [27]. Also, findings from cross-cultural research make similar claims, showing that individuals in societies that differ along the tightness-looseness dimension (i.e., the strength of social norms and the tolerance of deviance) manage their relationships differently in terms of interpersonal trust. Individuals in tight societies, compared with loose ones, tend to rely on strongly defined norms that provide clear expectations

enforced through external control systems [28,29], while those in loose cultures, where rules and regulations are less prevalent, are more likely to rely on interpersonal trust instead [30].

On the other hand, a different theoretical perspective considers formal institutions as laying the ground for the development of interpersonal trust, especially when there is high level of uncertainty about others' beliefs and norms [31]. Accordingly, institutions initially enforce trustworthy interactions through external rules, but after repeated successful interactions, individuals would not rely on mere assurance anymore, generalizing their beliefs about others' benevolence to other settings [32]. In fact, evidence shows that societies with efficient institutions, operationalized for example as societies with high level of democracy or effectiveness in enforcing agreements between strangers, display greater levels of interpersonal trust, as compared to countries with inefficient institutions [7,9].

## The relationship between institutional and interpersonal trust

Importantly, together with the quality of the institutions, individuals' perceptions and subjective assessments based on existing information of institutions might play a major role in understanding how institutions affect trust and the underlying mechanisms. Indeed, existing empirical evidence shows that, compared to institutional quality indicators, perceptions of institutions are similarly associated to interpersonal trust. For example, citizens' perceptions of institutional fairness or perceptions of corruption are related to interpersonal trust as do country-level indicators of fairness (e.g., skewness of income distribution) and corruption (e.g., number of arrests for corruption) [33,34]. Thus, it is possible to assume that, although in some situations external institutions may undermine interpersonal trust, this effect is conditional to whether people perceive them as legitimate and trustworthy (e.g., [35,36]). That is, taking into account whether these institutional constraints are trusted themselves can be crucial for understanding how they affect interpersonal trust. Indeed, institutional trust, defined as the extent to which individuals accept and perceive institutions as benevolent, competent, reliable, and responsible toward citizens [37], has been proposed as especially relevant for sustaining interpersonal trust [38,39]. For individuals, trusting institutions involves the perception that institutions would act in accordance to the common interest and society's needs to resolve disputes [40,41]. Recent evidence drawn from survey studies suggests that countries with high institutional trust are also characterized by more interpersonal trust [42,43]. This correlation appears robust across different countries and institutional settings, e.g., in Europe [44], Asia [45,46], and the USA [47], suggesting that these two types of trust are interrelated, with institutional trust influencing interpersonal trust. Given the observational nature of these studies, some authors argue that this relationship might actually be reversed (i.e., with interpersonal trust influencing institutional performance; e.g., [48]) or of mutual influence (e.g., [49]), while other more recent studies with individual fixed effects and cross-lagged panel models suggest that this is unlikely, and that institutional trust has an impact on interpersonal trust (e.g., [44,50–52]). Institutional trust has also been proposed as a mediator mechanism to explain how quality of institutions relates to interpersonal trust. For example, in a recent study, Lo Iacono showed that the ineffective institutions mostly affected interpersonal trust via a decrease in institutional trust [38]. Similarly, in a pilot study we found that institutional trust mediated the relationship between the presence (vs. absence) of institutions and trusting beliefs and behavioral intentions (methods and results are presented in detail in S1 Appendix in S1 File).

Although the existence of a relationship between institutional trust and interpersonal trust has been extensively discussed (e.g., [39]), empirical evidence to illuminate the underlying processes is scarce. In a pioneering experiment, Rothstein and Eek [53] manipulated corruption with scenarios describing corruption of institutional representatives in a fictitious country in

order to test the effect of institutions on interpersonal trust. In this study, student participants from Sweden and Romania were exposed to eight vignettes which respectively manipulated a bribe (present or absent), the initiator (the authority or the citizen), and the outcome of the exchange (positive or negative). The results showed that when public authorities were depicted as corrupt, participants perceived fellow citizens in the scenario as less trustworthy. The authors of the study further speculated that this effect could be explained by an inference-based underlying mechanism. Accordingly, individuals would make generalized inferences about others' trustworthiness based on observations of corrupt behavior enacted by public officials, others, and even themselves, interpreting these signals as information about what type of "game" is being played in a society [53,54]. Up to date, this remains the only experimental evidence available. However, the authors did not test the mechanisms underlying this effect, nor did they test the implications for trusting behavior. As such, little is still known about the question of whether institutional trust influences interpersonal trust beliefs and behavior toward unrelated strangers, or what underlying processes are responsible for this relationship. The present studies were designed to fill this void and to provide preliminary evidence for a potential underlying mechanism that might inform this debate.

## The mediating role of feelings of security

Previous literature suggests that, among their other functions, institutions have a crucial role for individuals to achieve security and safety in life [55,56]. Benevolent institutions provide structure in society, which allows individuals to achieve a greater sense of control over their lives [57]. Indeed, individuals are motivated to avoid victimization and exploitation from others [58] and to pursue safety and security by reducing personal threats in social situations [59]. Accordingly, institutions related to law and order may lead citizens to experience generalized feelings of security that would make them feel protected from potential offenses perpetrated by other fellow citizens [60]. We define these feelings as individuals' generalized perception of how safe they feel and to what extent they feel protected from socially threatening events. Additionally, we propose that when challenged by threats, feelings of security may be influenced by cues of trustworthiness of formal institutions, such as encounters with corrupt public officials or witnessing corrupt exchanges between other citizens and public representatives, as they provide a strong signal that social order is not guaranteed [53,60]. When it comes to interpersonal trust, there is evidence that general emotional states play a role [61], but, in particular, feelings of security are relevant to build interpersonal trust because they lead individuals to feel less vulnerable, which is key in trusting interactions with unknown strangers [14].

Law and order institutions are specifically related with protecting one's personal safety, but also social institutions [56], religious institutions [62], or the government [12] have been proposed to serve this motive and affect individuals' feelings of security. Despite the underlying role of feelings of security has never been investigated in previous research, other indirect empirical evidence is also in line with this assumption. In experiments, individuals prefer to interact in settings where sanctioning and rewarding institutions are in place [63]. These institutionalized societal models are particularly effective in mitigating people's fear of exploitation and, thus, to establish a culture of cooperation over time [64]. Further evidence in this vein comes from survey research, suggesting that efficient institutions promote less parochial behavior and more interpersonal trust [55]. Moreover, recent survey data support the protective function of trust toward governmental and legal institutions in Sweden focusing on fear of crime [12], showing that crime-related insecurity mediated the relationship between institutional trust and interpersonal trust.

## The current studies

The aim of the current three studies is to provide a first experimental and cross-country test for the hypothesis that institutional trust indirectly promotes trust among strangers by providing feelings of security, which in turn allow people to accept vulnerability and to trust others. Fig 1 summarizes the model tested across all studies, as well as the expected relationships between the constructs. In Studies 1 to 3, we tested whether feelings of security mediate the relationship between institutional and interpersonal trust. In Study 1, through a survey, we investigated whether trust in several formal institutions is related to feelings of security, and subsequently, to interpersonal trust. In Study 2, we tested our model with a multilevel mediation analysis on European Social Survey data (ESS; [65]) across 16 countries. Finally, in Study 3, we addressed the same hypothesis by directly manipulating institutional trust in a between-subjects experimental design. Additionally, in Study 3 we also test an alternative mediation model, according to which institutional trust affects interpersonal trust through an increase of the expectations about other's behavior. For an overview of the different operationalizations of the constructs used to test the model across all studies, and their descriptive statistics, see S1 Table in S1 File.

To establish the robustness of our results, all studies included control variables to account for variation in the dependent measure. Indeed, the lack of control for stable psychological dispositions in previous cross-sectional studies might have overestimated the relationship between the two forms of trust in the past (see [39]). In Studies 1 and 3, we controlled for individual dispositions related to interpersonal and institutional trust, namely trust propensity, political orientation, and security values. Trust propensity is a stable individual disposition, defined as a generalized expectation about others' trustworthiness, and one of the most significant predictors of trust in interactions with strangers [13,66]. This variable was included in Study 1 and 3 to disentangle that trusting beliefs and behavior toward a specific target did not depend on an underlying general willingness to trust others. Right-wing political orientation and, more generally, conservative ideology [67] may both be associated with the need to reduce uncertainty and support of external control systems [68]. The endorsement of security values characterizes individuals who prioritize security and predictability [37] and, therefore,

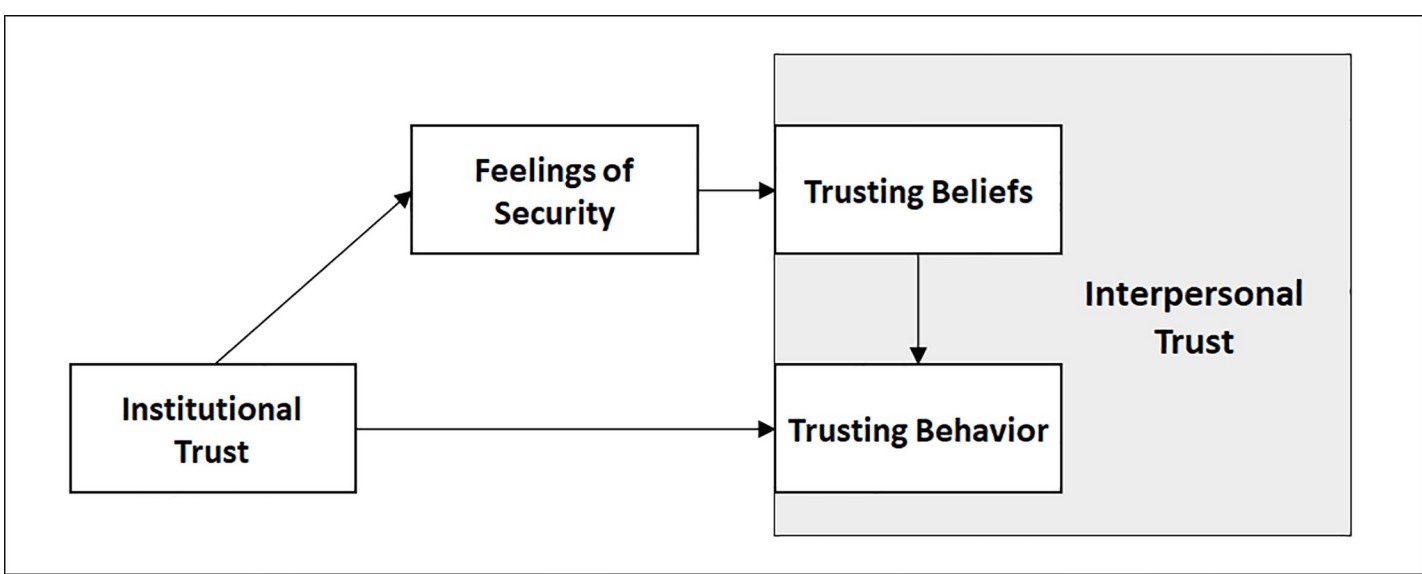

**Fig 1. The model: Institutional trust as a predictor of interpersonal trust (trusting beliefs and subsequent behavior) via feelings of security.**

are more likely to prefer strong institutions. In Study 2, we included country-level indicators of institutional performance that strongly correlate with both institutional and interpersonal trust (e.g., political and economic performance; [69]).

## Study 1

In Study 1, we aimed to extend findings on the relationship between institutional trust and interpersonal trust by providing a preliminary first test of the hypothesis that institutional trust promotes interpersonal trust toward strangers by enhancing individuals' feelings of security. In doing so, we used a cross-sectional survey design that analyzed trusting beliefs toward state institutions and fellow citizens. Additionally, we assessed institutional trust toward a large set of public institutions to identify whether the perceiving feelings of security would be relevant for specific institutions (such as those absolving more monitoring or sanctioning functions) or generalizable across institutional settings. Unlike previous observational studies that use single items to measure institutional trust, institutional trust was measured through scales covering perception of competence, benevolence, and integrity [70], to overcome single-items psychometric limitations (e.g., random measurement errors and biases in interpretation).

## Materials and methods

### Participants

The sample consisted of 181 Italian participants (75.7% female; $M_{age}$ = 28.06, $SD_{age}$ = 9.74). Most of them had bachelor's degree (40.3%) and described themselves as slightly left-wing on a 10-point political orientation scale ranging from left to right ($M$ = 4.06, $SD$ = 2.31). The participants' regions of origin were proportionally distributed among north (43.1%), center (14.9%), and south Italy (42%). To ensure that all participants had some degree of experience or previous information about Italian public institutions, we recruited participants being at least 18 years of age, and excluded participants that reported to have a different nationality ($N$ = 1). Sensitivity analysis revealed that this sample size would result in 80% statistical power to detect a small effect of institutional trust on trust beliefs ($f^2$ = 0.04; [71]). The whole research was conducted in accordance with the Declaration of Helsinki (7th revision, 2013) and local ethical guidelines for experimentation with human participants and was approved by the institutional review board at the University of Turin and by the ethical commission of the Zeppelin University in Friedrichshafen. All participants gave written informed consent prior to the experiments.

### Procedure

Participants were recruited on social media through a snowball sampling. They accessed to the online survey through a call for participation posted on social media accounts and were requested to share the call with others. The study included measures of trust toward different state institutions, interpersonal trust, feelings of security enhanced by public institutions, security values, trust propensity, and a socio-demographic section. To avoid sequence effects, all items were presented in a randomized order within each scale and, unless otherwise stated, they were answered on a seven-point Likert scale from 1 (*I completely disagree*) to 7 (*I completely agree*).

**Institutional trust.**  We assessed trust in five different institutions related to social order ([70], eight items for each institution, e.g., *"I trust the police in Italy because they behave benevolently toward citizens"*, police: α = .81; legal system: α = .84; government: α = .81; media: α =

.82; religious institutions: $\alpha = .87$). Following Agroskin, Jonas, and Traut-Mattausch [72], in addition to the items proposed in the original scale, we presented an additional item to increase the scale validity (i.e., "*I generally trust the police in my country*"). Following previous research on institutional trust (e.g., [73]), we then additionally averaged the scores of these five scales in a single cumulative index of institutional trust, which showed a good reliability, $\alpha = .70$.

**Interpersonal trust.** Trusting beliefs toward Italian citizens were measured through an adaptation of General Trust Scale ([25]; six items, e.g., "*I believe that Italian citizens are basically honest*", $\alpha = .93$). This scale consisted in a series of questions that assessed to what extent respondents perceived fellow citizens as trustworthy.

**Feelings of security.** We included a measure of feelings of security that individuals experience in relation to institutional performance and representatives (three items, i.e., "*I feel protected by public institutions*", "*I am comforted by thinking that I can count on public institutions if anything happens to me*", "*I feel I can rely on public institutions to assert my own rights*", $\alpha = .89$).

**Control variables.** As control variables, we measured trust propensity through the Trust in Others Scale ([74]; three items, e.g., "*I dare to put my fate in the hands of most other people*", $\alpha = .65$) and the endorsement of security values using the respective subscale from the Portrait Values Questionnaire [75]. Here respondents are asked to indicate own perceived similarity to a described person (five items, e.g., "*It is important to him to live in secure surroundings. He avoids anything that might endanger his safety*", $\alpha = .74$) on a six-point Likert scale from 1 (*not like me at all*) to 6 (*very much like me*).

## Results

Results showed the expected significant associations between average and distinct trust in the five institutions, feelings of security, and trusting beliefs toward citizens (S2 Table in S1 File). Additionally, the individual dispositions used as controls confirmed the hypothesized association with both interpersonal trust and institutional trust. The Pearson's correlation analyses also showed associations between trust propensity and interpersonal trust ($r = .39$, $p < .001$), the endorsement of security values and institutional trust ($r = .22$, $p = .003$), and right-wing political orientation and trust toward the police ($r = .27$, $p < .001$) and religious institutions ($r = .21$, $p = .006$), respectively. We also found a medium correlation between institutional trust and feelings of security ($r = .50$, $p < .001$), suggesting a possible partial overlap between the two constructs. However, confirmatory factor analyses showed that this is unlikely, and that security and institutional trust are separate factors (statistical details are reported in Table A in S2 Text in S1 File).

**The main effect of institutional trust on interpersonal trust.** We conducted a simple linear regression with institutional trust (average trust in all five institutions) predicting trusting beliefs. Institutional trust significantly predicted trust toward fellow citizens (interpersonal trust), $F(1, 179) = 15.29$, $p < .001$, $R^2 = .08$.

**The indirect effect of institutional trust on interpersonal trust via feelings of security.** A mediation analyses with 5,000 bootstrapped samples using the SPSS macro Process model 4 [76] showed that the indirect effect of institutional trust on trusting beliefs towards other citizens is significant for each of the five institutions under analysis: police ($b = 0.12$; 95% CI [0.05; 0.21], $R^2 = .12$), legal system ($b = 0.14$; 95% CI [0.06; 0.24], $R^2 = .11$), government ($b = 0.13$; 95% CI [0.06; 0.23], $R^2 = .12$), religious institutions ($b = 0.04$; 95% CI [0.01; 0.09], $R^2 = .02$), and the media ($b = 0.09$; 95% CI [0.04; 0.16], $R^2 = .12$). Overall, the relationship between institutional trust and trusting beliefs towards fellow citizens was mediated by feelings of

security while considering the aggregate index of institutional trust as a predictor in the model, $b = 0.18$; 95% CI [0.06; 0.32], $R^2 = .13$. These results appeared robust even if trust propensity, security values, political orientation, and education were included as controls in the analyses, $b = 0.15$; 95% CI [0.05; 0.27], $R^2 = .26$. Detailed results of the mediation models are presented in detail in Table 1.

## Discussion

The findings of Study 1 are in line with previous studies and our key hypothesis. Institutional trust was associated with interpersonal trust, and this relationship was mediated by the feeling that institutions can protect individuals from the exploitative behavior of strangers. This

**Table 1. Results of mediation models for the effect of institutional trust (aggregate measure and single institutions) on interpersonal trust through feelings of security.**

| Predictor and effect | Outcome: Trusting beliefs | | | | | |
|---|---|---|---|---|---|---|
| | Model 1 | | | Model 2 | | |
| | *b* | *SE* | 95% CI | *b* | *SE* | 95% CI |
| Mediator: Feelings of security | 0.23 | 0.07 | 0.08; 0.37 | 0.19 | 0.07 | 0.05; 0.33 |
| *Predictor: Institutional trust (Aggregate)* | | | | | | |
| Total effect | 0.39 | 0.10 | 0.19; 0.59 | 0.29 | 0.10 | 0.10; 0.49 |
| Direct effect | 0.21 | 0.11 | -0.01; 0.44 | 0.15 | 0.11 | -0.07; 0.37 |
| Indirect effect | 0.18 | 0.06 | 0.06; 0.32 | 0.15 | 0.06 | 0.06; 0.27 |
| Mediator: Feelings of security | 0.25 | 0.07 | 0.11; 0.39 | 0.19 | 0.07 | 0.06; 0.33 |
| *Predictor: Institutional trust (Police)* | | | | | | |
| Total effect | 0.25 | 0.07 | 0.10; 0.40 | 0.22 | 0.08 | 0.07; 0.36 |
| Direct effect | 0.13 | 0.08 | -0.03; 0.29 | 0.11 | 0.08 | -0.06; 0.27 |
| Indirect effect | 0.12 | 0.04 | 0.05; 0.21 | 0.11 | 0.04 | 0.04; 0.20 |
| Mediator: Feelings of security | 0.26 | 0.07 | 0.12; 0.41 | 0.2 | 0.07 | 0.06; 0.34 |
| *Predictor: Institutional trust (Legal system)* | | | | | | |
| Total effect | 0.22 | 0.07 | 0.08; 0.36 | 0.18 | 0.07 | 0.05; 0.31 |
| Direct effect | 0.08 | 0.08 | -0.08; 0.23 | 0.07 | 0.08 | -0.08; 0.22 |
| Indirect effect | 0.14 | 0.05 | 0.06; 0.24 | 0.11 | 0.04 | 0.03; 0.19 |
| Mediator: Feelings of security | 0.26 | 0.07 | 0.12; 0.40 | 0.20 | 0.07 | 0.07; 0.34 |
| *Predictor: Institutional trust (Government)* | | | | | | |
| Total effect | 0.24 | 0.08 | 0.09; 0.39 | 0.19 | 0.07 | 0.04; 0.33 |
| Direct effect | 0.11 | 0.08 | -0.06; 0.27 | 0.09 | 0.08 | -0.07; 0.24 |
| Indirect effect | 0.13 | 0.04 | 0.06; 0.23 | 0.10 | 0.03 | 0.04; 0.18 |
| Mediator: Feelings of security | 0.28 | 0.06 | 0.16; 0.41 | 0.23 | 0.06 | 0.11; 0.35 |
| *Predictor: Institutional trust (Religious institutions)* | | | | | | |
| Total effect | 0.12 | 0.06 | 0.01; 0.25 | 0.06 | 0.06 | -0.06; 0.18 |
| Direct effect | 0.09 | 0.06 | -0.03; 0.20 | 0.03 | 0.06 | -0.09; 0.15 |
| Indirect effect | 0.04 | 0.02 | 0.01; 0.09 | 0.03 | 0.02 | -0.01; 0.07 |
| Mediator: Feelings of security | 0.27 | 0.07 | 0.14; 0.40 | 0.22 | 0.06 | 0.10; 0.35 |
| *Predictor: Institutional trust (Media)* | | | | | | |
| Total effect | 0.21 | 0.08 | 0.06; 0.37 | 0.14 | 0.08 | -0.01; 0.30 |
| Direct effect | 0.13 | 0.08 | -0.03; 0.28 | 0.07 | 0.08 | -0.9; 0.22 |
| Indirect effect | 0.09 | 0.03 | 0.04; 0.16 | 0.07 | 0.03 | 0.03; 0.14 |

Results based on 181 observations. Model 1: Mediation analyses did not include control variables. Model 2: Mediation analyses included trust propensity, security values, political orientation, and education as control variables.

indirect effect appeared to be independent from individual dispositions. Moreover, Study 1 showed that this indirect effect was consistent over all the five main public institutions related to social order, even for those that do not directly deal with monitoring or sanctioning (e.g., the media). However, religious institutions seemed to play a more marginal role, as suggested by the magnitude of the effect size and the fact that the indirect effect becomes non-significant after control variables were added.

## Study 2

Study 1 provided initial evidence for the hypothesis that institutions, when trusted, are associated with feelings of security, which in turn predict trust in strangers. However, our results were limited to relatively small and non-representative samples from a single country and institutional context (i.e., Italy). In Study 2, we filled this gap by testing this hypothesis across 16 countries. Importantly, the current study presents different operationalizations of feelings of security and interpersonal trust. Feelings of security were measured as perception of personal safety. This measure is not explicitly tied to the specific institutions under investigation as in Study 1, but it is particularly relevant for institutions that are supposed to regulate social exchange and safeguard law and order. A different operationalization of feelings of security also allows to further generalize the findings and disentangle any possible overlap with the construct of institutional trust. Interpersonal trust has been assessed using respondents' scores on the Generalized Trust Scale. Differently from Study 1, the current trust measure does not involve a specific target of the trusting beliefs, but rather reflects beliefs toward "most people", that are likely to drive behavior in contexts involving unfamiliar actors. This different operationalization of interpersonal trust allows to relate our findings to previous evidence from survey studies using this scale (e.g., [43]) and to generalize them above specific trust targets (i.e., members of own community such as Italian citizens in Study 1). Finally, we included several control variables specifically related to both respondents' trust-relevant socio-demographic characteristics and institutional performance.

## Materials and methods

### Participants and procedure

This study used data from the European Social Survey (ESS). In total, answers from 180,051 participants (50.65% female; $M_{age}$ = 47.27, $SD_{age}$ = 18.04) from 16 countries of the European area (Belgium, Switzerland, Germany, Denmark, Spain, Finland, France, United Kingdom, Hungary, Ireland, Netherlands, Norway, Poland, Portugal, Sweden, and Slovenia) were included in the analysis. Most respondents completed upper secondary education (35.7%) and described themselves as politically moderate ($M$ = 5.08, $SD$ = 2.11) on a bipolar 11-point scale from 0 (*left-wing*) to 10 (*right-wing*). ESS data are gathered through cross-sectional face-to-face individual interviews administered to nationally representative samples, with different samples recruited for each wave. To provide greater temporal stability of the theoretical assumption, we decided to include only those countries participating to all survey waves (1–7) from 2002 to 2014, as well as only respondents without missing values in the variables described below (initial and final sample sizes and main descriptive statistics for each country are reported in S3 Table in S1 File). For the present analysis, we selected measures of trust in different state institutions, interpersonal trust, feelings of security, and country level indicators of institutional performance.

**Institutional trust.**    As in Study 1, we obtained the measure of institutional trust by aggregating trust ratings toward four state institutions (i.e., "*How much you personally trust the parliament*", "*the legal system*", "*the police*", and "*the politicians*", α range among countries = .78 -

.87). Individual responses were given on a 11-points Likert scale, ranged from 0 (*no trust at all*) to 10 (*complete trust*).

**Feelings of security.** The study contained a single-item measure as a proxy for respondents' feelings of security (i.e., "*How safe do you—or would you—feel walking alone in this area after dark*?"). Answers have been reverse-scored to allow consistent interpretation with the three-items measure used in Study 1, thus ranging from 1 (*very unsafe*) to 4 (*very safe*).

**Interpersonal trust.** Interpersonal trust was assessed using the Generalized Trust Scale (three items, e.g., "*Generally speaking, would you say that most people can be trusted, or that you can't be too careful in dealing with people*?", α range = .63 - .77). As for institutional trust, each item was answered on a 11-points Likert scale from 0 (*no trust at all*) to 10 (*complete trust*), phrased according to the specific item from 0 (*you can't be too careful*) to 10 (*most people can be trusted*).

**Control variables.** Last, country-level indicators of institutional performance of political and economic institutions were retrieved from established databases for each of the countries under investigation (i.e., government effectiveness, political rights, rule of law, economy competitiveness, GINI, and GDP per capita) from available time points between wave 1 and 7 of the ESS (2002–2014). A detailed description of these indicators is provided in S2 Appendix in S1 File.

## Results

First, data were corrected for sampling errors, since the ESS data involved respondents from multiple countries. By applying the design weight variable (*dweight*) included in the ESS dataset, we adjusted for the differences in the chance of selection of respondents for each country in all the following analyses. Moreover, control variables were included in the models in two steps, with respondents' trust-relevant socio-demographic characteristics (i.e., age, gender, and education) being included first, followed by institutional performance indicators at a second stage. We report results from both analyses.

**The indirect effect of institutional trust on interpersonal trust via feelings of security.** To test our main prediction that institutional trust predicts interpersonal trust through increased feelings of security, we performed multilevel mediation analyses with bootstrapping method with the R package mediation [77,78] to account for the nested structure of the ESS data. In this model, respondents (level-1) are nested within countries (level-2). The mediation functions took as an input two multilevel regression models. The first multilevel regression had institutional trust as independent variable and feelings of security as dependent measure with countries as random intercept. The independent variables of the second multilevel regression were institutional trust and feelings of security, while the dependent variable was interpersonal trust, again with country as random intercept.

In line with the hypothesis, the results showed that the feelings of security had a significant indirect effect in interpersonal trust, $b = 0.013$, 95% CI [0.0117; 0.0136] controlling for survey wave and common sociodemographic variables associated to interpersonal trust in survey research (i.e., gender, age, and education; e.g., [79]). The relationship was partially mediated since the effect of institutional trust on interpersonal trust, $b = 0.342$; 95% CI [0.3377; 0.3458], remained significant when the mediator was included in the model, $b = 0.329$; 95% CI [0.3251; 0.3325]. Moreover, the results showed that including institutional performance indicators (e.g., GDP per capita, government effectiveness, and rule of law) as covariates in the multilevel mediational analysis did not affect the significance of the model, $b = 0.013$; 95% CI [0.0123; 0.0145]. While controlling for institutional performance indicators, the effect of institutional trust on interpersonal trust, $b = 0.352$; 95% CI [0.3484; 0.3563], remained significant when the

**Table 2. Results of the multilevel mediation models for the effect of institutional trust (aggregate measure) on interpersonal trust through feelings of security.**

| Predictor and effect | Model 1 | | Model 2 | | Model 3 | |
|---|---|---|---|---|---|---|
| | *b* | 95% CI | *b* | 95% CI | *b* | 95% CI |
| Mediator: Feelings of security | 0.298 | 0.2978; 0.2984 | 0.326 | 0.3260; 0.3265 | 0.296 | 0.2954; 0.2962 |
| *Predictor*: Institutional trust (Aggregate) | | | | | | |
| Total effect | 0.352 | 0.3479; 0.3564 | 0.342 | 0.3377; 0.3458 | 0.352 | 0.3484; 0.3563 |
| Direct effect | 0.339 | 0.3347; 0.3421 | 0.329 | 0.3251; 0.3325 | 0.339 | 0.3354; 0.3424 |
| Indirect effect | 0.013 | 0.0121; 0.0144 | 0.013 | 0.0117; 0.0136 | 0.013 | 0.0123; 0.0145 |
| % of Total effect | 0.04 | | 0.04 | | 0.04 | |

Results of all models are based on 180,051 observations and use countries as random effects. % of Total effect: Proportion mediated (i.e., ratio of the total effect to the indirect effect).

Model 1: Mediation analyses included survey wave as control variable. Model 2: Mediation analyses included survey wave and individual-level variables (gender, age, and education) as control. Model 3: Mediation analyses included survey wave and country-level variables (political rights, government effectiveness, rule of law, economy competitiveness, GINI coefficient, and GDP per capita) as control.

mediator was included in the model, $b = 0.339$; 95% CI [0.3354; 0.3424]. Results of the multilevel mediation models are presented in detail in Table 2.

As in Study 1, we also run our models considering trust in the four different institutions, instead of a single aggregated measure. As expected, results showed consistent patterns, suggesting that the effect can be generalized over a variety of institutions. The details of such analyses can be found in S4 Table in S1 File. Given that estimates of indirect effects were similar across the four institutions ($b$ range = 0.007–0.011), we limit the report to trust in the police, as it will be the focus of Study 3. Overall, the relationship between trust in the police and interpersonal trust was mediated by feelings of security, $b = 0.007$; 95% CI [0.0068; 0.0087], even controlling for individual differences $b = 0.008$; 95% CI [0.0076; 0.0091], and differences in the quality of institutions, $b = 0.008$, 95% [0.0069; 0.0084].

As an additional robustness check, we also tested our hypotheses with a different model specification. We included countries as fixed effects, thus removing the multilevel structure. The results (using both aggregate measure of institutional trust and trust in the four different institutions) were in line with those presented above, even controlling for individual differences (see S5 Table in S1 File).

## Discussion

The results of Study 2 replicated findings of Study 1, using cross-national data with representative samples. Taking together responses obtained from participants of 16 different countries, we found that individuals who trusted institutions tended to feel more secure, which resulted in higher levels of interpersonal trust. Remarkably, these effects have been observed even considering more general feelings of security, not directly tied to the specific institutions as in Study 1. Moreover, by including country-level indicators of institutional performance (both political and economic), we found support for the hypothesis that the indirect effect of the feelings of security is related more to individuals' perception of the public institutions, rather than their actual efficiency and performance.

## Study 3

Study 1 and 2 provided support for the hypothesis that institutional trust is positively associated to interpersonal trust, and that the effect is related to the feelings of security that institutions convey. In Study 3, we aimed to experimentally test this mechanism by manipulating

institutional trust, in order to replicate the results obtained in Study 1 and 2. The manipulation consisted in providing participants with specific information about the institution's competence, benevolence, and reliability, in order to elicit consequent trust assessments. Additionally, we wanted to extend our claims on trusting beliefs to actual trusting behavior in an economic game. Thus, we tested our main prediction that institutional trust promotes trusting behavior, operationalized by investments in a trust game with a stranger, by increasing feelings of security conveyed by institutions.

Moreover, we also tested an alternative mechanism, to disentangle a possible explanation that the effect of institutions merely depends on the expectations regarding the behavior of the interaction partner. This would allow to test the hypothesis that interpersonal trust is influenced by feelings of security rather than a change in normative expectations brought about by formal institutions.

## Materials and methods

### Participants

A total of 94 participants (70.2% female; $M_{age}$ = 25.45, $SD_{age}$ = 6.24) were recruited from an online panel (i.e., Sona System) at a large Austrian University and received a 2 € show-up fee and a behavior-depending remuneration. Most had a high school diploma (44.7%) and reported a moderately left-wing political orientation ($M$ = 4.12, $SD$ = 1.57). Participants were mainly Austrians (59.5%), 22.3% were Germans, and the remaining 18.1% were German-speakers from other countries. Sensitivity analysis revealed that this sample size would result in 80% statistical power to detect a medium effect of institutional trust on trusting behavior ($d$ = 0.58; [71]).

### Procedure

Participants were invited to take part in an incentivized online study on decision-making. They learned that their choices could be matched with those of other anonymous participants from ten different countries, whose identity or belonging country would not be disclosed at any time. This matching protocol was introduced to manipulate institutional trust, providing the respondents with different information about the police in the partner's home country. After the manipulation, we measured participants' trusting beliefs toward the partner, trusting behavior, and expectations of reciprocity. Then, they completed a questionnaire assessing the feelings of security enhanced by the police depicted in the scenario, trust propensity, security values, risk attitudes, and political orientation.

**Experimental manipulation (institutional trust).** Past research found that individuals lacking perfect information about others (e.g., in interactions with strangers) make inferences about others' trustworthiness based on behavior of public officials in that society [53]. Following the same approach, at the beginning of the study, participants were randomly assigned to one of the two experimental conditions manipulating institutional trust (low vs. high) and were exposed to two scenarios providing different information about the police in the trustee's home country (referred to as Country X from now on), that was never disclosed across the entire duration of the study. They read a fictitious report of a survey about police's performance and perception in Country X the previous year. Following the definition of institutional trust as perception of benevolence, competence, and reliability of public institutions toward citizens (e.g., [37]), in the *low institutional trust* condition, the police were depicted as poorly qualified, neither fulfilling their obligations and nor serving the collective interest. Conversely, in the *high institutional trust* condition, the police were described as extremely skilled, committed, and responsible (see S3 Appendix in S1 File).

**Manipulation checks.** Two measures were used as manipulation checks to evaluate the extent to which the scenario elicited different degrees of (low vs. high) trust in the police. First, we directly asked participants how much would they trust the police in Country X on a seven-point Likert scale from 1 (*not at all*) to 7 (*completely*). Additionally, participants were asked to guess the trustee's country out of a list of ten countries, select as those ranking highest (i.e., Switzerland, Luxembourg, Norway, Sweden, Finland) and lowest (i.e., Portugal, Slovenia, Hungary, Czech Republic, Greece) in institutional trust scores according to data from OECD [80]. We expected that participants exposed to the high institutional trust condition would have associated Country X to an actual country renowned for its institutional trust.

**Dependent variables (trusting beliefs and behavior) and mediating variables (feelings of security and expectations of reciprocity).** In this study, interpersonal trust was operationalized as trusting beliefs and trusting behavior (i.e., the money invested in a trust game played with an unknown other) [81]. Participants were endowed with 5 Lab Coins (LC), each worth € 0.30, and assigned to the role of trustor. At stage one of the game, the trustor could transfer any of this amount (0–5 LC) to the trustee, keeping the remaining for herself, being aware that the transferred amount would have been tripled by the experimenter while passing it to the trustee. In stage two, the trustee could have transferred any portion of the tripled amount received back to the trustor. Importantly, they were informed that their decision would be randomly matched with that of another participant in the pool who will play a complementary role at a later stage of data collection in order to determine the final payment from the game. In reality, all participants played the sole role of trustor, since we were only interested in trusting behavior. The use of deception in this study (i.e., providing participants with fictitious information about the institutions in place in the trustee's country) was functional to increase internal validity of the study and to set up a situation in which any observed effect on interpersonal trust would have been solely attributed to the manipulated information. Given that we did not assess actual return behavior, the final payment for the participants was determined by tripling the amount they transferred in the game (€ 0–1.50) and was paid in addition to the show-up fee. After verifying the comprehension of the game, we assessed expectation of reciprocity by asking to express the percentage of the money they expected to receive back (0–100). Then, as in Study 1, we measured trusting beliefs toward the trustee through an adaptation of General Trust Scale ([25], α = .88) and feelings of security using the 3-items measure (α = .96).

**Control variables.** To adjust for potential underlying baseline individual dispositions across the different conditions, a number of variables were included in the experimental design and analyzed as covariates in the mediational models. As Study 1, we included as covariates a measure of trust propensity ([74]; α = .75), security values ([75]; α = .72), and political orientation measured through a bipolar 10-point scale item. In addition, we included a measure of risk attitudes as further control, as trusting behavior consisted in an actual monetary investment decision, which is generally associated with individual risk preferences [82]. Risk attitudes were measured using the financial subscale of the Risk-Behavior Scale [83]. Here respondents are asked to indicate the likelihood to engage in a series of economically risky behaviors (10 items; e.g., "*Spending money impulsively without thinking about the consequences*", α = .68) on a seven-point Likert answering scale ranging from 1 (*extremely unlikely*) to 7 (*extremely likely*). To avoid sequence effects, all items were presented in a randomized order within each scale.

## Results

As expected, participants exposed to the scenario of high institutional trust reported more trust in the police of Country X ($M = 5.15$, $SD = 1.07$), compared to the other condition

**Table 3. Guessed trustee's country according to experimental condition.**

| Institutional Trust | Switzerland | Luxembourg | Norway | Sweden | Finland | Portugal | Slovenia | Hungary | Czech Republic | Greece |
|---|---|---|---|---|---|---|---|---|---|---|
| Low | 0 | 4.35% | 2.17% | 0 | 0 | 4.35% | 6.52% | 32.61% | 21.74% | 28.26% |
| High | 22.92% | 10.42% | 18.75% | 27.08% | 12.50% | 0 | 2.10% | 2.10% | 2.10% | 2.10% |

($M = 2.43$, $SD = 1.44$), $t(92) = -10.39$, $p < .001$, $d = 2.14$. Also, a Chi-square test was performed to test whether, consistently with the experimental condition, participants associated the description of Country X to an actual high trust vs. low trust country as classified in OECD official rankings. As predicted, those in the high institutional trust condition associated Country X to a high trusting country significantly more than in the other condition, $\chi^2(1, N = 94) = 68.12$, $p < .001$ (Table 3), suggesting that the manipulation was successful in eliciting the intended perception of partner's country institutions and, thus, institutional trust.

**The main effect of institutional trust on interpersonal trust.** To test whether institutional trust had a main effect on trusting beliefs and trusting behavior, we used an independent samples $t$ test comparing the trusting beliefs about the trustee and trusting behavior (i.e., the amount of money invested in the trust game) observed in the two experimental conditions. On average, participants transferred 70.6% ($SD = 26.7\%$) of their initial endowment to the trustee. Results of the $t$ test showed no differences between the two experimental groups in either trusting beliefs toward the trustee, $t(92) = -1.00$, $p = .317$, $R^2 = .01$, nor trusting behavior, $t(92) = -1.00$, $p = .321$, $R^2 = .01$ (see S6 Table in S1 File).

**The indirect effect of institutional trust on interpersonal trust via feelings of security.** A serial mediation (Process model 6) on a bootstrap sample of 5,000 participants showed a significant effect of institutional trust on the money invested in the trust game, mediated by the feelings of security, which impacted trusting beliefs toward the partner in the trust game, $b = 0.21$, 95% CI [0.06; 0.54], $R^2 = .11$. The full serial mediation model remained significant after controlling for trust propensity, security values, risk attitudes, political orientation, and education, $b = 0.19$, 95% CI [0.03; 0.54], $R^2 = .13$. Results of the mediations are presented in detail in Table 4.

**Testing two competing psychological explanations for enhanced interpersonal trust.** One additional intended contribution of Study 3 was to test whether the indirect effect of institutional trust on interpersonal trust could simply be explained by increased expectations of reciprocity rather than increased feelings of security. To examine this possibility, we conducted

**Table 4. Results of serial mediation models for the effect of institutional trust (manipulation) on interpersonal trust through feelings of security.**

| Predictor and effect | Outcome: trusting behavior | | | | | |
|---|---|---|---|---|---|---|
| | Model 1 | | | Model 2 | | |
| | *b* | SE | 95% CI | *b* | SE | 95% CI |
| Mediator 1: Feelings of security* | -0.07 | 0.10 | -0.28; 0.13 | -0.07 | 0.11 | -0.28; 0.15 |
| Mediator 2: Trusting beliefs* | 0.52 | 0.17 | 0.19; 0.85 | 0.51 | 0.18 | 0.15; 0.88 |
| *Institutional trust (Manipulation)* | | | | | | |
| Total effect | 0.28 | 0.28 | -0.27; 0.82 | 0.38 | 0.29 | -0.19; 0.95 |
| Direct effect | 0.34 | 0.34 | -0.34; 1.02 | 0.36 | 0.37 | -0.36; 1.09 |
| Indirect effect | 0.21 | 0.11 | 0.06; 0.54 | 0.19 | 0.12 | 0.03; 0.54 |

Results based on 94 observations. Model 1: Mediation analyses did not include control variables. Model 2: Mediation analyses included trust propensity, security values, political orientation, risk attitudes, and education as control variables.

*Estimates of regressions of the mediators (feelings of security and trusting beliefs, respectively) predicting trusting behavior.

a parallel mediation analysis (Process model 4) with 5,000 bootstrapped samples. The results show that feelings of security had a significant indirect effect on trusting beliefs toward the trustee, $b = 0.41$, 95% CI [0.13; 0.83], while expectations of reciprocity did not, $b = 0.02$, 95% CI [-0.01; 0.15] (all path coefficients are reported in S7 Table in S1 File).

## Discussion

The results of Study 3 replicated the indirect effect obtained in Study 1 and 2 with experimental data. Institutional trust enhanced feelings of security, which in turn significantly predicted interpersonal trust. As expected, the extent to which participants transferred money to the other person depended on their achieved feelings of security from the institutions. Study 3 also generalized these effects on trusting behavior with real incentives, providing evidence for the external validity of the results.

Differently from Study 1 and 2, we did not observe a main effect of institutional trust on trusting beliefs or behavior. One possibility to explain these different findings can be attributable to the operationalization of institutional trust. In Study 3, institutional trust was indirectly manipulated by providing information about institutional performance, aimed at creating different perceptions of institutions in the two experimental condition. In the two surveys, however, respondents' perceptions of institutions were built across years of repeated exposure to institutional performance and behavior of institutional representatives.

## General discussion

Interpersonal trust among strangers is key for the societal, political, and economic development of a state. Given that interactions are embedded in a context regulated by societal institutions, recent research has increasingly focused on the questions whether and how these institutions can enhance (or impair) the development of interpersonal trust. In three studies, we tested the hypothesis that institutional trust indirectly promotes trust among strangers by increasing feelings of security, which in turn allow people to accept vulnerability and to trust others. Study 1 provided initial evidence for this underlying psychological process and showed that the association between institutional trust and interpersonal trust is mediated by feelings of security. Institutions that are trusted serve as a cue that individuals are protected, which in turn indirectly allows them to accept vulnerability and trust others. Study 2 further validated this initial evidence by analyzing cross-sectional data from 16 countries and using different operationalizations of the constructs. Finally, Study 3 provided an extension of the findings by manipulating institutional trust in an experimental design, and testing the effects on trusting behavior. In this study, we found again support for the indirect effect obtained in Study 1 and 2 with experimental data but, in contrast with these, we did not observe a main effect of institutional trust on trusting beliefs or behavior.

The current studies add an important piece to the puzzle on how micro-level perceptions and behaviors relate to macro-level societal processes. When it comes to institutional features, individual psychological processes often reflect the broader societal context. For example, individual self-regulation is higher in institutionally regulated countries [29], and citizens' intrinsic honesty is affected by country-level norm violations [84]. With respect to trust among strangers, our findings suggest that trusted institutions can provide feelings of security that serve as a basis to develop interpersonal trust [42,44]. Given the vulnerability to exploitation that trusting acts involve [14], we traced back the effect of institutional trust to the key functions of institutions to serve the fundamental need to feel protected. Indeed, trusting institutions does not automatically lead individuals to feel secure. Even if institutions are considered competent and reliable, individuals still may feel highly insecure in unpredictable and extreme situation (e.g.,

terroristic attacks, that highly increase distrust toward others). This is confirmed by our results, showing that the two constructs are only moderately correlated (Pearson's $r$ = .50 and $r$ = .18 in Study 1 and 2, respectively) and load on different factors.

Our findings are not in line with research that would propose that institutions have a detrimental effect on interpersonal trust in light of their primary role of providing assurance by affecting the incentive structure and individuals' normative expectations [25,26]. In Study 2, we included a series of control variables related to institutional quality to control for the possibility that the actual performance of institutions is driving the effect. Among those, there were estimates of the ability of government to provide high quality public services, implement effective policies, protect legal entitlements, and to maintain social order through formal rules [85]. These variables did not affect the significance of the model. Moreover, when institutional trust was manipulated in Study 3, we observed an indirect effect on trusting behavior even if the institution was not directly involved in the interaction and had no actual power to influence the outcome of the interaction and provide assurance (e.g., by punishing exploitative actions). Also, this study sought to rule out that this effect could be merely explained by positive expectations about the partner's behavior. Although this was not a key goal of the research, this finding suggests that social inferences are unlikely to be the main mechanism underlying the relationship between institutional and interpersonal trust [54]. If that would have been the case, we should have observed expectations to mediate this relationship, given that institutional representatives (the police in Study 3), according to this approach, should act as a signal of the type of game played in a certain society. Future research should further explore this question by designing ad hoc studies to understand the relative contribution of feelings of security and social inference-based mechanisms.

The current findings highlight the need to actually implement cues that generate trust in public institutions such as transparent communication, legitimate law enforcement, or explicit anti-corruption policies to promote trust in the general society and thereby, social and economic prosperity [46]. If public institutions cannot fulfil the need of individuals to feel secure, citizens may turn away from established institutions, start to individually protect themselves (e.g., by investing in lawyers and insurances), and support or develop new parallel institutions (e.g., alternative media) to achieve this need and restore trust levels. In extreme cases, as response to perceived insecurity, individuals might start to endorse nationalist positions [86], and even turn to anti-social organizations like the mafia in order to restore their lack of security [87,88]. Also, the results have implications for policy in light of research on the effects of individuals' past experiences of victimization on interpersonal trust. In particular, trust has been showed to be hardly affected by direct victimization [89], but rather by a more general fear of crime, developed as response to the environmental context, such as segregated and disadvantaged neighborhoods [12,79]. This can undermine efficacy of interventions exclusively aimed at reducing victimization experiences (such as severe monitoring and sanctioning of violent and property crimes). Thus, public institutions might combine these interventions together with others with a focus on restoring feelings of security, such as the implementation of direct and accessible dialogue and transparent communication with citizens, as well as the possibility to monitor openly this process, especially in disadvantaged contexts where fear of victimization is more prevalent.

## Limitations and future research

Our findings have some limitations that need to be acknowledged. Due to the use of observational data across Study 1 and 2, it may be argued that our results are affected by endogeneity. That said, we addressed this issue in two ways. First, in both Study 1 and 2 we included

relevant control variables related to stable individual dispositions to prevent overestimating the hypothesized relationships, which are often overlooked in current research practices (see [39]). Second, we specifically integrated the survey evidence provided in Study 1 and 2 with an experiment that exogenously manipulated institutional trust by providing specific information about institutions in the trustee's country (Study 3).

Another potential limitation is the lack of a main effect of institutional trust in Study 3, which was observed in Study 1 and 2. That said, this finding does not affect the overall conclusion we derive from the results for several reasons. First, the main focus of this work was testing the indirect effect of institutional trust on interpersonal trust and this effect was replicated across all studies (with different datasets and operationalizations), even when tested against competing mechanisms. Moreover, our manipulation in Study 3 was carefully designed from previous research which found a significant effect on trusting beliefs toward strangers based on the behavior of institutional representatives [53]. Therefore, it is possible that higher sample sizes would be successful in detecting a significant main effect. Although previous research found evidence that the presence of cues (or inferences), rather than the actual implementation of mechanisms (such as reputation), is enough to elicit an effect [90], future research might consider to identify the boundary conditions of this effect by designing a setting that more closely resembles daily experience with (not) trusted institutions, that could actually intervene in the situation (e.g., punishing untrustworthy behavior), and not involving the use of deception, in order to draw stronger conclusions about whether and how this perception, and subsequent feelings, affect interpersonal trust.

Moreover, even if we collected evidence from several countries, all respondents of the current studies were from Western democratic countries in which institutional trust levels are generally high. Additionally, two of the three studies involved convenience samples and, thus, not representative of the national population. Thus, a remaining question for the future would be to test these hypotheses on a more diverse set of institutional contexts and to test the generalizability in non-Western societies. Last, although exchanges with strangers within a society often do not exceed a single interaction, future research could explore the role of institutional trust in repeated interactions, where reputational information comes into play.

All things considered, the present findings appear robust and generalizable across research methodologies and variables operationalizations, and remain consistent even when controlling for relevant individual characteristics and institutional performance indicators. Study 1 allows to generalize the results across the entire set of studies by firstly testing the reliability of the three-items measure of feelings of security and trusting beliefs then adopted in Study 3, and to provide a more fine-grained measure of institutional trust as compared to the standard single-item questions adopted by the ESS in Study 2. Additionally, by testing our hypotheses across different conceptualizations of strangers (i.e., fellow citizens, most people, a person from a foreign country), the current set of studies controlled for the risk of refer to different targets, ranging from family members to people from other nationalities, while answering to the standard interpersonal trust question that is widely diffused in survey research (i.e., trust radius problem; [91]).

## Concluding remarks

Trusting strangers is a fundamental pillar of human societies. Understanding how institutional trust can shape and maintain interpersonal trust, with a focus on individual's needs, brings together converging insights from different research traditions and methodologies. The present work provides survey, cross-cultural, and experimental evidence in support of the conclusion that trust in formal institutions is important for understanding variation in interpersonal

trust, especially when they manage to enhance a feeling of security among citizens. This is relevant both for the theoretical debate around interpersonal trust and social capital. But also, the role of institutions may become even more essential, as societies become more and more complex, and if anything, move away from small societies in which trust is largely based on tight groups who meet face-to-face in their community, the workplace, or the local café.

## Supporting information

**S1 File.**
(DOCX)

**S1 Data.**
(SAV)

**S2 Data.**
(SAV)

## Author Contributions

**Conceptualization:** Giuliana Spadaro, Katharina Gangl, Jan-Willem Van Prooijen, Paul A. M. Van Lange, Cristina O. Mosso.

**Data curation:** Giuliana Spadaro.

**Formal analysis:** Giuliana Spadaro.

**Investigation:** Cristina O. Mosso.

**Methodology:** Giuliana Spadaro, Katharina Gangl, Cristina O. Mosso.

**Supervision:** Katharina Gangl, Jan-Willem Van Prooijen, Paul A. M. Van Lange, Cristina O. Mosso.

**Writing – original draft:** Giuliana Spadaro, Katharina Gangl.

**Writing – review & editing:** Giuliana Spadaro, Katharina Gangl, Jan-Willem Van Prooijen, Paul A. M. Van Lange, Cristina O. Mosso.

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
