## [Decision Letter · Decision Letter 0]

8 Apr 2020

PONE-D-20-07477

Enhancing feelings of security: How trustworthy institutions promote interpersonal trust

PLOS ONE

Dear Dr. Spadaro,

Thank you for submitting your manuscript to PLOS ONE. After careful consideration, we feel that it has merit but does not fully meet PLOS ONE’s publication criteria as it currently stands. Therefore, we invite you to submit a revised version of the manuscript that addresses the points raised during the review process.

We would appreciate receiving your revised manuscript by May 23 2020 11:59PM. To enhance the reproducibility of your results, we recommend that if applicable you deposit your laboratory protocols in protocols.io, where a protocol can be assigned its own identifier (DOI) such that it can be cited independently in the future. For instructions see: http://journals.plos.org/plosone/s/submission-guidelines#loc-laboratory-protocols

We look forward to receiving your revised manuscript.

Kind regards,

Valerio Capraro

Academic Editor

PLOS ONE

Journal Requirements:

Additional Editor Comments (if provided):

I have now collected three reviews from three experts in the field. The reviewers are somehow split. One is very critical and suggests rejection. The other two are less critical, one suggests minor revision and the other one suggests major revision. After reading the manuscript myself, I have opted for following the majority and invite you to revise your work according to the reviewers' comments. Needless to say that all comments must be addressed to the best of your possibilities, including, and especially, those from the "negative" reviewer.

I am looking forward for the revision.

And I take this occasion to wish you and your loved ones to be safe and healthy during these difficult times.

Reviewers' comments:

Reviewer's Responses to Questions

**Comments to the Author**

1. Is the manuscript technically sound, and do the data support the conclusions?

Reviewer #1: No

Reviewer #2: Yes

Reviewer #3: Yes

2. Has the statistical analysis been performed appropriately and rigorously? 

Reviewer #1: No

Reviewer #2: Yes

Reviewer #3: Yes

3. Have the authors made all data underlying the findings in their manuscript fully available?

Reviewer #1: Yes

Reviewer #2: Yes

Reviewer #3: Yes

4. Is the manuscript presented in an intelligible fashion and written in standard English?

Reviewer #1: Yes

Reviewer #2: Yes

Reviewer #3: Yes

5. Review Comments to the Author

Reviewer #1: Thank you for the opportunity to review this piece. The authors take up a very interesting question regarding the oft-identified relation of trust in institutions and trust in individuals. As the authors rightly note, the question is a much contested one both in the direction of the effect and in its mechanism. The paper here purports to wade into this debate but fails to contribute for two major reasons.

1- Studies 1 and 2 add little new to this discussion (as the authors note, these are replications) but most concerningly, they are imprecise in measurement. The paper suggests that it takes Mayer and Rousseau's definitions of trust as their starting point but instead measure institutional trustworthiness. That measure of trustworthiness is then averaged across several institutions to create a poorly theorized amalgamation of trust in "institutions" with no attention to whether the individual has any real information about them. These are then connected with generalized trust in others which were also argued to be an important control variable that was missed in previous research. Thus, the DV, IV, and control for these studies all measure generalized trust or trustworthiness of a variety of targets in a way that makes them all just slightly different measures of trust propensity. As a result, the relations among them look like endogenity issues and little conceptual or empirical defense is offered to the contrary.

2- Study 3 seeks to identify a single institution and test the effect of changes in its trustworthiness on trust in a specific other. This helps a lot with the clarity lacking in S1 and S2 but S3 fails to find a significant direct effect of institutional trust on interpersonal. Nonetheless, the authors go on to test a mediation which does make me wonder if I missed something but the paragraph in the middle of page 26 pretty clearly states that "no differences emerged for either trusting beliefs... nor trusting behavior". Indeed, this makes a great deal of sense as I can't really see why hearing about the police in a random trust game partner's country would impact my thoughts about their behavior in the game. Nothing they are doing (returning or not returning the funds) would be illegal and it's asking a lot to think that a college student would think through the stimulus enough to believe that a truly effective police force would create a society in which people be more likely to return more money in a computer mediated trust game played as a way to get course credit. Without this direct effect, the rest of this study doesn't make a whole lot of sense to me.

Smaller concerns:

--I would encourage the authors to be a bit more careful in conceptualizing their constructs. Often the manuscript uses the terms trust, trustworthiness, or trusting behavior as interchangeable. I understand the strong empirical relations among the variables but this literature has progressed to a point where it's not hard to keep them separate. Indeed, the Mayer Davis and Schoorman article that was cited here is a major touch point for this clarity.

--I am a little concerned about the S1 and S3 samples' representativeness. Social media ads are often displayed according to algorithms and college student samples are notoriously non-generalizable, especially when thinking across cultures as this study wants to do. It may be that there is little to be concerned about here given the samples that were actually collected but more was needed to show that relations identified here could reasonably generalize.

--I would note that what has been randomly manipulated in this study is not, in fact, trustworthiness or trust, but instead information that is intended to impact those assessments. Manipulation checks help here as they can show whether--all else being equal (or at least randomly distributed)--trust assessments change as a result of the information presented. The first manipulation asks not about trust but likelihood of trusting (which may just be a translation issue) but the second has nothing to do with trust at all. Clearer defense of how the information that was randomly presented actually gets at the intended IV would be welcomed.

--I am pretty sure that PLoS ONE allows for in-text tables and figures. Relegating most of them to the appendix seems less than ideal.

Reviewer #2: [See attachment to visualize the review correctly]

This is, potentially, an interesting contribution to the literature on institutional and social trust. However, the manuscript has several issues and requires major revisions to be publishable. Here are some suggestions to improve the paper:

General

1. The authors argue that the relationship between social and political trust has been neglected in the literature (e.g. in the conclusions: “Decades of research have focused on several processes that may promote trust among strangers, but very little attention has been devoted on one recurrent feature that characterize modern human interactions: the presence of institutions”). However, as the authors acknowledge in their literature review, there is a relevant body of research investigating precisely this relationship following different approaches: Sonderskov, Brehm & Rahn, Lekti, Rothstein, Uslaner, Stolle have addressed this topic (all mentioned in the manuscript). I would suggest to add the following references as well:

Herreros F, Criado H. The state and the development of social trust. International Political Science Review. 2008;29(1):53–71

Lo Iacono S. (2019). Law-breaking, fairness, and generalized trust: The mediating role of trust in institutions. PloS one, 14(8).

Richey S. The impact of corruption on social trust. American Politics Research. 2010;38(4):676–90.

Js You. Social trust: Fairness matters more than homogeneity. Political Psychology. 2012;33(5):701–21.

I invite the authors to acknowledge previous research on institutional trust/institutions and social trust throughout the entire manuscript (in line with their literature review), while fleshing out more clearly the main contribution of the manuscript, namely the analysis of the mediation effect and the disentangling of the psychological processes behind the relationship (which, indeed, has not been empirically investigated, though theoretically argued to some extent – e.g. Rothstein and Stolle 2008).

2. The methods and results sections need extensive revising for all three studies. Given the wide variety of measures employed, it is often unclear how concepts are operationalized. A descriptive table showing the coding, the mean, SD, N of the variables used in each study would be extremely helpful (maybe you can include this in the SI). Also, it would be good to present results from the mediation models in a table (one for each study), showing the effects with and without covariates (in the SI you could report the complete tables with all coefficients). Side note: at p. 15 the authors mention that “the lack of control for individual differences in previous cross-sectional studies might have overestimated the relationship

between the two forms of trust in the past”. Looking at previous studies in the literature (see point 1), this is hardly correct (they do employ a wide variety of controls at the individual level). Also, while using controls in Studies 1 and 2 makes perfect sense because they are observational studies, Study 3 is an experiment and it shouldn’t require controls if the randomization worked properly. Including controls for Study 3 should be justified by arguing that you are adjusting for (potential) differences in baseline covariates across the different conditions.

3. There is no explanation or description of the pilot in the main text – the authors briefly mentioned it in the literature review at p. 11: “Thus, for a better understanding of the relation between institutions and interpersonal trust, in the next section, we will focus on research addressing the effect of institutional trust on interpersonal trust. In S1 Appendix we report the results of an additional experimental pilot study that manipulated the presence vs. absence of institutions, and provided evidence for its cascading effects on institutional trust, trusting beliefs, and trusting behavioral intentions towards a stranger”. I would suggest the authors to provide a more detailed justification of the pilot in the general description of their work.

4. The manuscript requires a careful revision of the text. Sometimes sentences appear quite disconnected, or are simply inconsistent with the rest of the paragraph. I report here a couple of cases, but an accurate double-check is required. Examples, p.15: “The whole research was conducted in accordance with the Declaration of Helsinki (7th revision, 2013) and local ethical guidelines for experimentation with human participants and was approved by the institutional review board at the University of Turin and by the ethical commission of the Zeppelin University in Friedrichshafen. All subjects gave written informed consent prior to the experiment. To avoid sequence effects, in Studies 1 and 3 all items were presented in a randomized order within each scale and, unless otherwise stated, they were answered on a seven-point Likert scale from 1 (I do not agree) to 7 (I totally agree)”.

At p. 17 (while describing Study 1): “From these items, we created a general aggregated measure of institutional trust by averaging scores of these five scales. As in Study 1, trusting beliefs toward Italian

citizens were measured through the adapted General Trust Scale ([23], α = .93)”.

5. In the manuscript, the authors often discuss the direct impact of institutional trust on interpersonal trust. Fig. 1, however, does not represent this accurately (as it shows only an effect on trusting behavior). Thus, I would suggest to modify Fig.1 and make it more consistent with the authors’ general argument/interpretation of results. Here is a possible “solution” (apologies for the sketchy picture):

[Fig.1 modified - see attachment]

Study 1

The contribution of Study 1 is difficult to grasp at the moment. Indeed, since it does not employ a representative sample of the population and there is no manipulation (not sure if it makes sense to report the results of the sensitivity analysis at p.16), Study 1 appears to contribute very little to the discussion, especially in comparison to Studies 2 and 3. I suggest to move Study 1 to the SI to give more space to Studies 2 and 3, which require more information. If the authors disagree with this comment, I believe they should better justify the value/input of Study 1 (i.e. how does this study exactly improve on our current knowledge of this relationship?). The following passage at p. 20 provides a good hint on where to start (from my point of view): “This different operationalization of interpersonal trust would allow to relate our findings to previous evidence from survey studies using this scale (e.g., [36]) and to generalize them above specific trust targets (i.e., members of own community such as Italian citizens in Study 1)”.

Study 2

1. Given that the mediation mechanism does not involve level 2 (i.e. country-level) variables and they are not interested in exploring the impact of any country-level variables on trust, I don’t see why the authors use multilevel mediation. Instead, they could simply control for all country differences (i.e. having countries as a fixed-effect, as they do for survey-years). This should tell us whether the mediation effect is working across countries, which would be more relevant/interesting for their analysis. If the mediation effect is not working once they control for countries, then it might be interesting to understand why this is the case/for which countries the mediation works/what country-level factors are important in this respect (re-introducing the multi-level mediation here).

2. Considering that in Study 3 the authors are manipulating trust towards the police (rather than the broader concept of institutional trust), it would be good to have a separate part of the analysis focusing on the same aspect in Study 2 (trust in the police � feelings of security � generalized trust). This would also help the reader to see more clearly the link between the two studies.

3. As mentioned in the general comments section, I believe that Study 2 needs a table where you concisely present the results from the mediation model (i.e. the different paths for direct and indirect effects) with and without covariates (e.g. Model 1 no controls, Model 2 controls at the individual level, Model 3 controls for countries and survey-years).

Study 3

1. Study 3 requires a more detailed presentation of the design (e.g. how many sessions did you have? How many people per session? It would be good to have more details on the trust game – one-shot? Strategy method? How much could the trusters send? What was the multiplier? Etc.). At the moment, it is difficult to understand what the subjects exactly experienced and in which order (e.g. did they play the trust game after the questions on trusting beliefs? Did the questions on expectations of reciprocity followed the trust game?). This is important to properly evaluate the results of the study.

2. The type of behavioural trust you are measuring here is quite different from the one measured in Studies 1 and 2. Indeed, here subjects are asked whether they would trust someone from another country (i.e. Country X – trustee’s home country. Side note: it would be good to know why you had those 11 countries, on which basis you selected them etc.). This is not equivalent to measuring trust towards unknown fellow citizens/strangers, as the form of trust measured in Study 3 involves a stronger out-group component (it should be closer to trust towards migrants). The theoretical framework of Study 3 should discuss this issue and interpret findings accordingly.

3. In my view, deception could have been avoided (by designing the experiment more carefully). I would invite the authors to justify their decision, explaining why deception was needed in this case.

4. Recoding of trusting behavior in Table S4 does not seem consistent with results presented at p.26: “On average, participants transferred 70.6% (SD = 26.7%) of their initial endowment to the trustee, […] (see S4 Table)”. While in table S4 you report the following:

[See attachment]

Institutional Trust

Low High

M (SD) M (SD)

Trusting behavior 3.39 (1.39) 3.67(1.28)

How should we interpret the values 3.39 or 3.67 in relation to the value of 70.6%? As mentioned in the general comments, the coding of variables is quite confusing, and it appears to be inconsistent in some passages. Please double-check carefully the manuscript in this respect.

5. Table 2 was cut and didn’t properly show the results.

6. I would like to invite the authors to elaborate more on the mediation effect reported in Study 3. Indeed, while there is no significant direct effect from the manipulation of trust (towards the police) to trusting behaviors, the analysis suggests that there is a significant indirect effect (through feelings of insecurity). Is this a case of indirect-only mediation (e.g. Zhao, X., Lynch Jr, J. G., & Chen, Q. 2010. Reconsidering Baron and Kenny: Myths and truths about mediation analysis.)? Or is it actually due to a moderating effect? Also, how is this consistent with results reported in Study 2 where, in my understanding, we have both significant direct and indirect effects? How do the authors explain this difference? How should we interpret findings from Studies 2 and 3 once taken together?

Reviewer #3: 1) I think the paper could cite the existing and quite large literature in economics on the relationship between institutions and trust, in particular the work by Luigi Guiso and Guido Tabellini.

2) I am not sure about the value added of Study 1. Study 2 also uses observational data and therefore suffers from the same problem as Study 1, but is superior and more powerful given the larger number of observations and the different countries. In my view, Study 2 makes Study 1 redundant. If the authors decide on keeping Study 1, I encourage them to point out more clearly the differences to Study 2, and in particular what we learn from Study 1 that we do not learn from Study 2.

3) Statistical Analysis: Overall, the statistical analysis is conducted appropriately, but I have some questions:

- Do you include country or subnational region fixed effects in Study 2? They would absorb many fixed institutional and economic characteristics of a country or region that could affect institutional trust and trust in strangers.

- For the reader it would be interesting to know for each study what is the percentage of the effect of total effect of institutional trust on trust that is mediated by safety considerations.

- The t-test you mention is not clear to me ("On average, participants transferred 70.6% (SD = 26.7%) of their initial endowment to the trustee, and no differences emerged for either trusting beliefs... " Do you test the difference in money transferred between the two experimental groups? What is the result?

4) Presentation of results:

- I find the tables not easy readable and not very intuitive. For example, variable names are not intuitive, number of observations are missing, and it is not clear from the table what are the control variables included in the regressions.

- There is no Table with results for Study 2.

- Table 2 that contains the results for Study 3 is too large and half of it is not readable.

6. PLOS authors have the option to publish the peer review history of their article (what does this mean?). If published, this will include your full peer review and any attached files.

Reviewer #1: No

Reviewer #2: No

Reviewer #3: No

---

## [Author Response · Author response to Decision Letter 0]

5 Jun 2020

Dear Dr. Valerio Capraro, 

Enclosed is a revised version of our manuscript “Enhancing feelings of security: How trustworthy institutions promote interpersonal trust” which we submitted for publication in PLOS ONE (Manuscript PONE-D-20-07477). We were grateful to receive three extremely constructive reviews, each of which expressed enthusiasm about the potential of the paper, as well as constructive criticism and comments for improving the paper. 

We greatly appreciate your and the reviewers’ constructive comments on our previous manuscript. We have thoroughly considered (and sometimes discussed) each of these suggestions and used them to strengthen the manuscript. Below we respond in detail to each of your own and the reviewers’ comments. For an easier navigation of the manuscript, our responses are listed in italics, and when changes have been made to the manuscript, we insert a section of text from the paper and/or provide the page numbers of the tracked version of the revised manuscript where the revisions can be found. We hope you find this structure easy to navigate. Thanks again for this opportunity and we look forward to hearing about your decision.

Reviewer #1

Thank you for the opportunity to review this piece. The authors take up a very interesting question regarding the oft-identified relation of trust in institutions and trust in individuals. As the authors rightly note, the question is a much contested one both in the direction of the effect and in its mechanism. The paper here purports to wade into this debate but fails to contribute for two major reasons.

1- Studies 1 and 2 add little new to this discussion (as the authors note, these are replications) but most concerningly, they are imprecise in measurement. 

Thank you for raising the concern about the magnitude of the contribution of Study 1 and 2 to the current debate. Although part of the research questions has been addressed in the past, in our view (see below) is that our set of studies contribute new insights to the literature. But your comment helped us realize that we did not adequately discuss or illustrate the novelty of our findings in the previous version of the manuscript (as Reviewers 2 and 3 pointed out as well). 

The main contribution of both studies is to test the mediational role of feelings of security, an underlying mechanism that has been suggested as potentially relevant in previous research (e.g., Robbins, 2011; Rothstein & Stolle, 2008) but that has never been empirically tested. We revised the introduction section (p. 7) and the introduction to Study 1 (p. 9) to make this contribution more explicit. Moreover, although the relation between institutional and interpersonal trust has been already investigated, we believe that being able to replicate previous research provided us a solid base on which then we could confidently add incremental insights about the underlying psychological mechanism. This seems especially important in times when replicability of research in psychology is a serious issue and often discussed within and beyond the boundaries of the scientific literature. 

We also made sure to enhance clarity and precision of all aspects regarding definition, measurement, and operationalization of the key constructs in the new version of the manuscript.

 Please see below our responses regarding your measurement concern.

The paper suggests that it takes Mayer and Rousseau's definitions of trust as their starting point but instead measure institutional trustworthiness. 

Thank you for this point. This gives us a chance to clarify our choices in terms of measurement and how they relate to the literature. The focus of this research is on the interplay of trust in institutions and trust in strangers. We cite pioneering work by Mayer et al. (1995) and Rousseau et al. (1998) in defining interpersonal trust, our outcome variable as “a psychological state that involves the intention to accept vulnerability in social interactions, under conditions of social risk and interdependence” (p. 2), as trust toward strangers involves the highest risk and uncertainty.

However, it is important to note that the definition of institutional trust (our predictor variable in the model) is rooted in a different (interdisciplinary) literature. Specifically, institutional trust is defined as “the extent to which individuals accept and perceive institutions as benevolent, competent, reliable, and responsible toward citizens” (p. 5). The measures used to operationalize institutional trust reflects this individual perception in Study 1 (e.g., “I trust the police in Italy because they behave benevolently toward citizens”) and Study 2 (e.g., “How much do you personally trust each of the following institutions”), respectively. We edited and re-structured the method section of all studies to provide more clarity. Throughout the manuscript, we double-checked all terms we used to ensure to use always exactly the same wording instead of variations. Thanks to your comment, our terminology became more stringent now.

That measure of trustworthiness is then averaged across several institutions to create a poorly theorized amalgamation of trust in "institutions" with no attention to whether the individual has any real information about them.

Our approach in selecting multiple different institutions had the aim to gain insights on the generalizability of our assumption, thus was guided by our research question, and the aggregation of institutional trust assessments across a range of institutions is a common practice adopted in previous research in institutional trust (e.g., Irwin, 2009; Nannestad et al., 2014; Seifert, 2018; Zmerli & Newton, 2008).

As we describe in the introduction, we assumed that feelings of security being mostly conveyed by public institutions that deal with law and order (p. 6-7), but previous literature suggests that a broad range of institutions might fulfill this function (e.g., Mayseless & Popper, 2007) as long as they convey a sense of protection. Accordingly, we did not limit our investigation to one particular institution and, indeed, our findings support the generalizability of this effect across different institutions.

That said, we see the benefit of not exclusively performing the analyses at the aggregated level. For this reason, together with models using the aggregated index of institutional trust as a predictor, we now report the results of a series of additional analyses (with and without control variables) using the single institutions in Study 1 (Table 1) and Study 2 (S4 Table and S5 Table). Overall, all analyses remained significant when trust in a single institution was considered as predictor (the only exception was for the indirect effect of religious institutions, which became non-significant after controlling for individual differences, which is discussed at p. 14).

As to your point about making sure that individuals have any real information about institutions, in Study 1 our approach to making sure that all respondents had information about the institutions of the country was to only select respondents with at least 18 years of age and with Italian nationality, which are the prerequisites to exercise voting rights in Italy. We now report this in the text (p. 10).

These are then connected with generalized trust in others which were also argued to be an important control variable that was missed in previous research. Thus, the DV, IV, and control for these studies all measure generalized trust or trustworthiness of a variety of targets in a way that makes them all just slightly different measures of trust propensity. As a result, the relations among them look like endogenity issues and little conceptual or empirical defense is offered to the contrary.

Our main goal is to understand the relation between institutional trust and how this can affect trust toward strangers. We agree that these different forms of trust may be potentially explained by an underlying propensity to trust others (defined by Mayer et al. as “a stable within-party factor that will affect the likelihood the party will trust. [...]. Propensity will influence how much trust one has for a trustee prior to data on that particular party being available”, p. 715). This is the reason why we considered it important to include a measure of trust propensity in Studies 1 and 3. This measure is an often-used as general form of readiness to trust, aimed to capture differences in general trust between people, rather than differences in trust among specific partners; such partner-specificity is therefore not included in the measurement. In our view, it is reassuring that we find support for our hypotheses whether or not we control for individual differences in trust propensity.

The inclusion of a trait variable that can account for the variation in the outcome is a common practice, and has been applied also in the trust field. For example, the same theoretical model of Mayer and colleagues include both trust and trust propensity, and previous research tested hypotheses accounting for multiple types of trust in the same model (e.g., institutional trust, trust propensity, and trusting beliefs in Moin et al., 2015). We hope that with the current revision we clarified this even further.

Regarding possible endogeneity in the model, we believe that combining observational evidence (Study 1 and 2) with experimental (Study 3) provides compelling evidence about the robustness of the hypothesized relations. In particular, we designed Study 3 with the explicit goal to implement an exogenous manipulation of institutional trust and found evidence for the proposed indirect effect (i.e., institutional trust impacting trust in others via feelings of security). This evidence, along with past research that has disentangled the directionality of the relation between trust in institutions and trust in others, provide evidence in support of the convergent validity of the present findings. We specifically discuss this aspect in the “Limitations and future research” section of the revised version (p. 30). 

2- Study 3 seeks to identify a single institution and test the effect of changes in its trustworthiness on trust in a specific other. This helps a lot with the clarity lacking in S1 and S2 but S3 fails to find a significant direct effect of institutional trust on interpersonal. Nonetheless, the authors go on to test a mediation which does make me wonder if I missed something but the paragraph in the middle of page 26 pretty clearly states that "no differences emerged for either trusting beliefs... nor trusting behavior". 

Indeed, this makes a great deal of sense as I can't really see why hearing about the police in a random trust game partner's country would impact my thoughts about their behavior in the game. Nothing they are doing (returning or not returning the funds) would be illegal and it's asking a lot to think that a college student would think through the stimulus enough to believe that a truly effective police force would create a society in which people be more likely to return more money in a computer mediated trust game played as a way to get course credit. Without this direct effect, the rest of this study doesn't make a whole lot of sense to me.

Thanks for this point. You are correct in pointing out that the results did not reveal a main effect of our manipulation on interpersonal trust (operationalized as both trusting beliefs and behavior). 

First of all, in our model we predicted both a main and an indirect effect (see opening of “The current studies” p. 8). Even if we did not find a main effect, we find a significant indirect effect via feelings of security and trusting beliefs. To do so, we performed a path analysis in light of the large consensus that the lack of a main effect should not be considered as a necessary prerequisite to perform such a test, especially when the relationships among the variables are theoretically guided (Shrout & Bolger, 2002; Zhao et al., 2010). Moreover, we tested this path model against a competing model and found no support for the latter.

That said, we completely agree with you that this only provides support for part of our hypotheses and that we should discuss the lack of main effect. Following your comment, we also revised the manuscript to relate findings of Study 3 to those obtained in Studies 1 and 2 and provided possible explanations for the observed lack of main effect (p. 26-27, p. 30).

For example, regarding your point on the manipulation, we think that our rationale is justified for two main reasons. First, we designed our manipulation to affect interpersonal trust in light of what has been proposed by Rothstein & Eek (2009). That is, individuals lacking perfect information about others (e.g., in interactions with strangers) make inferences about others’ trustworthiness based on behavior of public officials in that society. This guided our choice of the manipulation (i.e., providing information about performance of the police). In the current version, we clarify the rationale of the manipulation (p. 22).

Second, we have reasons to believe that even if we did not implement such a setting in which institution could directly regulate the social exchange, the current design of Study 3 is suitable to test our hypotheses about interpersonal trust. Research in psychology has found that mere cues of reputation, or group membership can have an effect on behavior even in one-shot interactions (Romano et al., 2017; Yamagishi & Mifune, 2008), where no actual future consequences should not matter (as they would in repeated interactions). We revised the introduction (p. 4-5) to make clear that our predictor variable is not related to actual performance of institutions (such as in Herreros & Criado, 2008; Lo Iacono, 2019)

As to the reason why we did not observe such effect in place, there are two potential possibilities that we now discuss in the General Discussion section. First, it is true that in our manipulation of information about police in partner’s country the institution does not have a direct link to the game (i.e., there are no formal rules or laws broken in the trust game and, even if, the police cannot actually intervene to regulate the social exchange). This might be a reason why we did not observe a main effect. Testing these hypotheses in an experimental setting that more closely resembles daily experience with trustworthy – or untrustworthy – institutions, that could actually intervene in the situation (e.g., punishing untrustworthy behavior) is an interesting direction for future research, which we now discuss at page 30-31. Another possibility is that people just learned about the institution and were asked a one-shot interaction. Future research may also address this aspect, by investigating whether the effect would emerge in repeated interactions (with both institution and the partner). 

As a minor note, following your comment about the “computer mediated trust game played as a way to get course credit”, we disambiguated the description of the incentive structure in the text to clarify that decisions were economically incentivized (both as show-up fee and behavior-dependent remuneration) and course credits were not at stake (p. 23).

Smaller concerns:

--I would encourage the authors to be a bit more careful in conceptualizing their constructs. Often the manuscript uses the terms trust, trustworthiness, or trusting behavior as interchangeable. I understand the strong empirical relations among the variables but this literature has progressed to a point where it's not hard to keep them separate. Indeed, the Mayer Davis and Schoorman article that was cited here is a major touch point for this clarity.

We realized that, indeed, in the previous version of the manuscript we used some inconsistent wording that could have caused some confusion. We removed all the inconsistencies and now only stick to institutional trust, trusting beliefs, trusting behavior, and trust propensity. Importantly, in light of these changes, we decided to edit the title of the paper, as the previous version mentioned “trustworthy institutions”. The original wording could have been a primary source of confusion as the focus of the research is institutional trust. Also, following the excellent suggestion of Reviewer 2, we created a new table (S1 Table) that summarizes the operationalization, definition, and descriptive statistics for each variable in the model. This overview should be helpful, as these measures are rooted in somewhat different literatures (even disciplines).

--I am a little concerned about the S1 and S3 samples' representativeness. Social media ads are often displayed according to algorithms and college student samples are notoriously non-generalizable, especially when thinking across cultures as this study wants to do. 

This is a fair point. But allow us to clarify what we initially defined as “social media posting” in Study 1. The call for participation was not displayed as an advertisement (thus, was not affected by algorithms in targeting specific populations), but as a regular post coming from one of our social media accounts, that was shared with more users with a snowball sampling. We specify this in the text now (p. 10).

Moreover, please also note that Study 2 was a large-scale study including representative data from 16 countries, and yielded findings that were consistent with the other results. We believe that this alleviates this concern substantially.

It may be that there is little to be concerned about here given the samples that were actually collected but more was needed to show that relations identified here could reasonably generalize.

The samples used in Study 1 and 3 are not representative and resemble more the type of samples that are the norm in the psychological field. However, we also agree with the importance of generalizability and we tested our model on European Social Survey data which are nationally representative in Study 2, finding consisting results. In Study 1, we collected a sample that, although not representative, was distributed across the Italian regions (“The participants’ regions of origin were proportionally distributed among north (43.1%), center (14.9%), and south (42%) Italy”, p. 10). In Study 3, the sample is perfectly comparable to the sample examined in the original experiment of Rothstein & Eek (2009), and on which current experimental evidence is based.

Nevertheless, we now acknowledged the non-representativeness issue of Studies 1 and 3 in the limitation section and suggest it as next step in the future directions of this line of research (p. 31).

--I would note that what has been randomly manipulated in this study is not, in fact, trustworthiness or trust, but instead information that is intended to impact those assessments. Manipulation checks help here as they can show whether--all else being equal (or at least randomly distributed)--trust assessments change as a result of the information presented. The first manipulation asks not about trust but likelihood of trusting (which may just be a translation issue) but the second has nothing to do with trust at all. Clearer defense of how the information that was randomly presented actually gets at the intended IV would be welcomed.

Thank you for the remark. We integrated new content that better clarifies how information about institutions relates to institutional trust, so that our manipulation in Study 3 would become more straightforward (p. 4 in the introduction, p. 20 in Study 3, and p. 30-31 in the final discussion). 

We believe that, once clarified that we are manipulating information intended to impact institutional trust, manipulation checks are adequate to assess whether we have been successful or not in eliciting such perceptions:

- For the first manipulation check, there might be a translation issue in place, as the original German wording was not intended to ask about “likelihood” of trusting (as hypothetical attitude, or as probability to trust). The response set should thus be interpreted as ranging from “to not at all trust” to “completely trust”. We disambiguated it and changed the text accordingly (p. 22).

- For the second manipulation check, we now provide more detail and rationale (p. 22). Also, we included the table showing the percentage of trustee’s countries guessed according to experimental condition (previously presented in the SI) in the main text for clarity (Table 3).

--I am pretty sure that PLoS ONE allows for in-text tables and figures. Relegating most of them to the appendix seems less than ideal.

In the updated version of the manuscript we revised the tables to include more information (Table 1 and 4) and included new tables (Table 2 and 3). 

Reviewer #2

This is, potentially, an interesting contribution to the literature on institutional and social trust. However, the manuscript has several issues and requires major revisions to be publishable. Here are some suggestions to improve the paper:

General

1. The authors argue that the relationship between social and political trust has been neglected in the literature (e.g. in the conclusions: “Decades of research have focused on several processes that may promote trust among strangers, but very little attention has been devoted on one recurrent feature that characterize modern human interactions: the presence of institutions”). However, as the authors acknowledge in their literature review, there is a relevant body of research investigating precisely this relationship following different approaches: Sonderskov, Brehm & Rahn, Lekti, Rothstein, Uslaner, Stolle have addressed this topic (all mentioned in the manuscript). I would suggest to add the following references as well:

Herreros F, Criado H. The state and the development of social trust. International Political Science Review. 2008;29(1):53–71

Lo Iacono S. (2019). Law-breaking, fairness, and generalized trust: The mediating role of trust in institutions. PloS one, 14(8).

Richey S. The impact of corruption on social trust. American Politics Research. 2010;38(4):676–90.

Js You. Social trust: Fairness matters more than homogeneity. Political Psychology. 2012;33(5):701–21.

I invite the authors to acknowledge previous research on institutional trust/institutions and social trust throughout the entire manuscript (in line with their literature review), while fleshing out more clearly the main contribution of the manuscript, namely the analysis of the mediation effect and the disentangling of the psychological processes behind the relationship (which, indeed, has not been empirically investigated, though theoretically argued to some extent – e.g. Rothstein and Stolle 2008).

Thank you for this suggestion and this list of relevant readings. They now are acknowledged in the literature review and helped us to better structure the introduction, and highlight how the present studies can contribute to this line of research. 

2. The methods and results sections need extensive revising for all three studies. Given the wide variety of measures employed, it is often unclear how concepts are operationalized. A descriptive table showing the coding, the mean, SD, N of the variables used in each study would be extremely helpful (maybe you can include this in the SI). 

We re-structured the methods sections of all three studies and created different sub-sections in which the variables are described and presented in a consistent order. As suggested also by Reviewer 1, Study 3 went through the most extensive re-writing to clarify each aspect of the experimental procedure. 

We also followed up on your suggestion and created a table that shows the variables used in each study (S1 Table) to test the model, together with their descriptive statistics information, a short definition and reference for each construct.

Also, it would be good to present results from the mediation models in a table (one for each study), showing the effects with and without covariates (in the SI you could report the complete tables with all coefficients).

Thanks for the suggestions, we have now included such tables (Table 1, 2, and 4) in the text. 

Side note: at p. 15 the authors mention that “the lack of control for individual differences in previous cross-sectional studies might have overestimated the relationship

between the two forms of trust in the past”. Looking at previous studies in the literature (see point 1), this is hardly correct (they do employ a wide variety of controls at the individual level). 

We agree that the studies we review in the introduction do employ many individual-levels control (e.g., age, gender, education, to mention a few). We originally used the term “individual differences” in the manuscript as used in the psychological literature (e.g., personality traits, attitudes, or values). We now changed this wording, and clarified this both in the introduction while discussing the control variables (p. 8) and in the overall discussion (p. 30). 

Also, while using controls in Studies 1 and 2 makes perfect sense because they are observational studies, Study 3 is an experiment and it shouldn’t require controls if the randomization worked properly. Including controls for Study 3 should be justified by arguing that you are adjusting for (potential) differences in baseline covariates across the different conditions.

We agree that the aim of the inclusion of control variables in Study 3 should be differentiated from the two observational studies. We have specified it in the “Control variables” section of Study 3 (p. 24).

3. There is no explanation or description of the pilot in the main text – the authors briefly mentioned it in the literature review at p. 11: “Thus, for a better understanding of the relation between institutions and interpersonal trust, in the next section, we will focus on research addressing the effect of institutional trust on interpersonal trust. In S1 Appendix we report the results of an additional experimental pilot study that manipulated the presence vs. absence of institutions, and provided evidence for its cascading effects on institutional trust, trusting beliefs, and trusting behavioral intentions towards a stranger”. I would suggest the authors to provide a more detailed justification of the pilot in the general description of their work.

In the revised manuscript, we introduced the pilot study differently and related it to existing literature (p. 5). We did not provide extensive detail about it as the tested model is different (i.e., institutional trust as a mediator and not as a predictor) and we did not want it to become a source of confusion. 

4. The manuscript requires a careful revision of the text. Sometimes sentences appear quite disconnected, or are simply inconsistent with the rest of the paragraph. I report here a couple of cases, but an accurate double-check is required. Examples, p.15: “The whole research was conducted in accordance with the Declaration of Helsinki (7th revision, 2013) and local ethical guidelines for experimentation with human participants and was approved by the institutional review board at the University of Turin and by the ethical commission of the Zeppelin University in Friedrichshafen. All subjects gave written informed consent prior to the experiment. To avoid sequence effects, in Studies 1 and 3 all items were presented in a randomized order within each scale and, unless otherwise stated, they were answered on a seven-point Likert scale from 1 (I do not agree) to 7 (I totally agree)”.

At p. 17 (while describing Study 1): “From these items, we created a general aggregated measure of institutional trust by averaging scores of these five scales. As in Study 1, trusting beliefs toward Italian

citizens were measured through the adapted General Trust Scale ([23], α = .93)”.

Thank you for this suggestion. We revised these and other sentences in the manuscript to improve the flow.

5. In the manuscript, the authors often discuss the direct impact of institutional trust on interpersonal trust. Fig. 1, however, does not represent this accurately (as it shows only an effect on trusting behavior). Thus, I would suggest to modify Fig.1 and make it more consistent with the authors’ general argument/interpretation of results. Here is a possible “solution” (apologies for the sketchy picture):

We agree that such an edit would make the figure more consistent with the model that has been tested across the three studies. We appreciated the proposed suggestion and your effort in providing us a possible solution. We adapted Fig. 1 accordingly.

Study 1

The contribution of Study 1 is difficult to grasp at the moment. Indeed, since it does not employ a representative sample of the population and there is no manipulation (not sure if it makes sense to report the results of the sensitivity analysis at p.16), Study 1 appears to contribute very little to the discussion, especially in comparison to Studies 2 and 3. I suggest to move Study 1 to the SI to give more space to Studies 2 and 3, which require more information. If the authors disagree with this comment, I believe they should better justify the value/input of Study 1 (i.e. how does this study exactly improve on our current knowledge of this relationship?). The following passage at p. 20 provides a good hint on where to start (from my point of view): “This different operationalization of interpersonal trust would allow to relate our findings to previous evidence from survey studies using this scale (e.g., [36]) and to generalize them above specific trust targets (i.e., members of own community such as Italian citizens in Study 1)”.

Thank you for this input. It gave us the chance to clarify the exact contribution of Study 1 to the entire set of studies (p. 9, p. 27). The main contribution we describe in the manuscript is to provide the possibility to generalize the results across the three studies, that used different operationalizations of the constructs:

- Study 1 adopted the 3-items measure of feelings of security for the first time and showed its reliability. This allowed to adopt the same measure in Study 3;

- Study 1 used the same measure of trusting beliefs adopted in Study 3, but toward a different target (i.e., Italian citizens and the partner in the trust game, respectively); 

- Study 1 measured institutional trust using scales that focus on competence, benevolence, and integrity of public institutions (Hofmann et al., 2014), which provides several advantages as compared to the standard “confidence in the following institutions” items (also adopted in the ESS).

We hope that, in light of the new information we added, we were able to convey why we think that the study should be discussed in the main text together with the evidence from the other studies. 

Study 2

1. Given that the mediation mechanism does not involve level 2 (i.e. country-level) variables and they are not interested in exploring the impact of any country-level variables on trust, I don’t see why the authors use multilevel mediation. Instead, they could simply control for all country differences (i.e. having countries as a fixed-effect, as they do for survey-years). This should tell us whether the mediation effect is working across countries, which would be more relevant/interesting for their analysis. If the mediation effect is not working once they control for countries, then it might be interesting to understand why this is the case/for which countries the mediation works/what country-level factors are important in this respect (re-introducing the multi-level mediation here).

Thank you for the suggestion. In the original analyses we used countries as random effects following previous research (e.g., Romano et al., 2017) but, indeed, the model specification you suggest could also be appropriate to test our hypotheses (as the involved variables are measured at the individual level). We now run these additional analyses and reported the mediation analyses with countries as control in the results section (p. 19), showing consisting results across the two model specifications. These results are presented in detail in S5 Table.

2. Considering that in Study 3 the authors are manipulating trust towards the police (rather than the broader concept of institutional trust), it would be good to have a separate part of the analysis focusing on the same aspect in Study 2 (trust in the police � feelings of security � generalized trust). This would also help the reader to see more clearly the link between the two studies.

We followed this suggestion and report the results using trust in the police as predictor in a separate section in Study 2 (p. 19). Additionally, in S4 Table and S5 Table we report the results for trust in all the single institutions taken separately. 

3. As mentioned in the general comments section, I believe that Study 2 needs a table where you concisely present the results from the mediation model (i.e. the different paths for direct and indirect effects) with and without covariates (e.g. Model 1 no controls, Model 2 controls at the individual level, Model 3 controls for countries and survey-years).

Done, this table is referred in the manuscript as S4 Table.

Study 3

1. Study 3 requires a more detailed presentation of the design (e.g. how many sessions did you have? How many people per session? It would be good to have more details on the trust game – one-shot? Strategy method? How much could the trusters send? What was the multiplier? Etc.). At the moment, it is difficult to understand what the subjects exactly experienced and in which order (e.g. did they play the trust game after the questions on trusting beliefs? Did the questions on expectations of reciprocity followed the trust game?). This is important to properly evaluate the results of the study.

We have now provided more detail about the game paradigm including endowment, multiplier, and detailed information on how financial rewards for participants have been calculated (p. 23).

Also, we have re-structured the procedure section to clarify the different stages of the process (including in which order we assessed all the variables). To this purpose, we now created four sections in the procedure, describing (1) Experimental manipulation (institutional trust), (2) Manipulation checks, (3) Dependent variables (trusting beliefs and behavior) and mediating variables (feelings of security and expectations of reciprocity), and (4) Control variables.

2. The type of behavioural trust you are measuring here is quite different from the one measured in Studies 1 and 2. Indeed, here subjects are asked whether they would trust someone from another country (i.e. Country X – trustee’s home country. Side note: it would be good to know why you had those 11 countries, on which basis you selected them etc.). This is not equivalent to measuring trust towards unknown fellow citizens/strangers, as the form of trust measured in Study 3 involves a stronger out-group component (it should be closer to trust towards migrants). The theoretical framework of Study 3 should discuss this issue and interpret findings accordingly.

We agree, and therefore now discuss how employing different trust target for the “strangers” would enhance our understanding of our evidence, mentioning that it allows the generalizability of the results in light of the trust radius problem (Delhey et al., 2011) (p. 27)

Additionally, we have now made more explicit the reason why we selected the specific countries (p. 22).

3. In my view, deception could have been avoided (by designing the experiment more carefully). I would invite the authors to justify their decision, explaining why deception was needed in this case.

We are happy to clarify. Although norms in psychological research are traditionally more lenient than other disciplines (e.g., experimental economics) when it comes to using deception, we agree that this should be avoided if the same experimental rigor can be obtained without it.

In this study, we aimed to provide an experimentally controlled manipulation of institutional trust by providing participants with very specific information about the institutions in place in the trustee’s country. To test such an effect on behavior for the first time, it was crucial to make sure to set up a clean situation in which any observed effect would be attributable to this randomly assigned information and not to any other pre-existing information about the partner. 

Accordingly, one way to avoid deception here would have involved running the studies across cultures (e.g., matching participants from a high vs a low trusting countries) and providing real statistics of institutional performance. However, at the time the research was conducted we did not have the resources to access to such participants pools. An alternative setting would have required to ask participants to make a hypothetical decision (i.e., behaving as if they would do in a real situation).

We now clarified in the manuscript which specific purpose served the use of deception in this study (p. 23) and encourage future studies to extend our findings without using it (p. 31).

4. Recoding of trusting behavior in Table S4 does not seem consistent with results presented at p.26: “On average, participants transferred 70.6% (SD = 26.7%) of their initial endowment to the trustee, […] (see S4 Table)”. While in table S4 you report the following:Institutional Trust

Low High

M (SD) M (SD)

Trusting behavior 3.39 (1.39) 3.67(1.28)

How should we interpret the values 3.39 or 3.67 in relation to the value of 70.6%? As mentioned in the general comments, the coding of variables is quite confusing, and it appears to be inconsistent in some passages. Please double-check carefully the manuscript in this respect.

Thank you for pointing this out. Indeed, we realized that the choice range (0-5 lab coins) was not provided in the original text. We extensively re-checked and revised this section (p. 23). We hope this information justifies how the 70.6% was calculated based on the means reported in the table. 

5. Table 2 was cut and didn’t properly show the results.

Our apologies if you could not access that information. The Plos One submission guidelines encourage to submit tables even if horizontally spread, as they could be seen and processed in “web layout”. However, we now submitted an auxiliary file in which all tables cited in the manuscript can be accessed and visualized in “print layout” (i.e., the default view for Word files) to facilitate the review process.

6. I would like to invite the authors to elaborate more on the mediation effect reported in Study 3. Indeed, while there is no significant direct effect from the manipulation of trust (towards the police) to trusting behaviors, the analysis suggests that there is a significant indirect effect (through feelings of insecurity). Is this a case of indirect-only mediation (e.g. Zhao, X., Lynch Jr, J. G., & Chen, Q. 2010. Reconsidering Baron and Kenny: Myths and truths about mediation analysis.)? Or is it actually due to a moderating effect? Also, how is this consistent with results reported in Study 2 where, in my understanding, we have both significant direct and indirect effects? How do the authors explain this difference? How should we interpret findings from Studies 2 and 3 once taken together?

Thanks for the opportunity to elaborate more on the effects reported in Study 3. Reviewer 1 similarly pointed out the lack of a main effect in that study. We carefully addressed this point in the revised version of the manuscript. Please see above our response to Reviewer 1’s remark.

Overall, we now discuss possible interpretations of the findings of the three studies taken together and we suggest directions for future research (p. 30-31) that might contribute to a further understanding of the phenomenon. 

As to your suggestion about an interaction effect being present, we do not think this is the case in the current study as we find that our experimental manipulation has a main effect on feeling of security (t = -7.793, p < .001) and that the indirect effect was consistent even after controlling for an independent and stable disposition (i.e., how much participants valued security in their life). Of course, there is the possibility that potential moderator variables might influence the relationship on top of this effect (e.g., a general need for security and protection, Mayseless & Popper, 2007). We think this is an interesting question, but it might deserve to be pursued in a separate project aiming at tackling such interaction.

Reviewer #3

1) I think the paper could cite the existing and quite large literature in economics on the relationship between institutions and trust, in particular the work by Luigi Guiso and Guido Tabellini.

We acknowledge the following additional references from the economic literature as particularly relevant for our literature review:

Guiso, L., Sapienza, P., & Zingales, L. (2016). Long-term persistence. Journal of the European Economic Association, 14(6), 1401-1436.

Tabellini, G. (2008). Institutions and culture. Journal of the European Economic Association, 6(2-3), 255-294.

Ljunge, M. (2014). Social capital and political institutions: Evidence that democracy fosters trust. Economics Letters, 122(1), 44-49.

2) I am not sure about the value added of Study 1. Study 2 also uses observational data and therefore suffers from the same problem as Study 1, but is superior and more powerful given the larger number of observations and the different countries. In my view, Study 2 makes Study 1 redundant. If the authors decide on keeping Study 1, I encourage them to point out more clearly the differences to Study 2, and in particular what we learn from Study 1 that we do not learn from Study 2.

Following your and Reviewer 2’s suggestions, we now clarify how exactly Study 1 contributes to our understanding of the relationship between interpersonal and institutional trust, and how it relates to Study 2 in this manuscript (p. 9, p. 27). You can find our response above under the Study 1 paragraph). Also, we now mention the value of having a different target of interpersonal trust (more oriented toward the ingroup members), in combination with the other targets of interpersonal trust (p. 27). 

3) Statistical Analysis: Overall, the statistical analysis is conducted appropriately, but I have some questions:

- Do you include country or subnational region fixed effects in Study 2? They would absorb many fixed institutional and economic characteristics of a country or region that could affect institutional trust and trust in strangers.

The analyses reported in the original manuscript did not include countries as fixed effects.

As a robustness check, and as suggested by Reviewer 2, we conducted additional analyses and tested our hypotheses with countries as fixed effects. We comment such analyses in the revised version of the manuscript (p. 19) and report the details in S5 Table)

- For the reader it would be interesting to know for each study what is the percentage of the effect of total effect of institutional trust on trust that is mediated by safety considerations.

We now report the percentage of the effect accounted for by the mediator in all mediation analyses performed in Study 3 (Table 2, S4 Table, S5 Table). However, we did not provide the information for Study 1 and 3, as literature suggests that the proportion mediated should be reported only when N ≥ 500 (Fairchild & McDaniel, 2017; MacKinnon et al., 1995) and therefore it could have raised interpretability concerns.

- The t-test you mention is not clear to me (“On average, participants transferred 70.6% (SD = 26.7%) of their initial endowment to the trustee, and no differences emerged for either trusting beliefs... “ Do you test the difference in money transferred between the two experimental groups? What is the result?

This is correct. We disambiguated this issue both extensively revising the procedure section (e.g., specifying the choice range 0-5) (p. 23), and the description of the t-test in the results (p. 25).

“Results of the t test showed no differences between the two experimental groups in either trusting beliefs toward the trustee, t(92) = -1.00, p = .317, R2 = .01, nor trusting behavior, t(92) = -1.00, p = .321, R2 = .01 (see S6 Table)”.

4) Presentation of results:

- I find the tables not easy readable and not very intuitive. For example, variable names are not intuitive, number of observations are missing, and it is not clear from the table what are the control variables included in the regressions.

- There is no Table with results for Study 2.

- Table 2 that contains the results for Study 3 is too large and half of it is not readable.

Thank you for the remarks. We now created new tables for all the three studies (Table 1, 2, 4) with more intuitive labels and additional information (i.e., path from the mediator to the outcome variable, total effect, direct effect, indirect effect). In the notes, we report which exactly the control variables were used for each model, together with the number of observations.

As for the readability of large tables, we apologized if you couldn’t have access to the full information while reviewing the manuscript. As we mentioned to Reviewer 2 who raised a similar point, we now submitted an auxiliary file in which all tables cited in the manuscript can be accessed and visualized to facilitate the review process.

References

Delhey, J., Newton, K., & Welzel, C. (2011). How general is trust in “most people”? Solving the radius of trust problem. American Sociological Review, 76(5), 786–807. https://doi.org/10.1177/0003122411420817

Fairchild, A. J., & McDaniel, H. L. (2017). Best (but oft-forgotten) practices: Mediation analysis. American Journal of Clinical Nutrition, 105(6), 1259–1271. https://doi.org/10.3945/ajcn.117.152546

Herreros, F., & Criado, H. (2008). The State and the development of social trust. International Political Science Review, 29(1), 53–71. https://doi.org/10.1177/0192512107083447

Hofmann, E., Gangl, K., Kirchler, E., & Stark, J. (2014). Enhancing tax compliance through coercive and legitimate power of tax authorities by concurrently diminishing or facilitating trust in tax authorities. Law and Policy, 36(3), 290–313. https://doi.org/10.1111/lapo.12021

Irwin, K. (2009). Prosocial behavior across cultures: The effects of institutional versus generalized trust. In S. R. Thye & E. J. Lawler (Eds.), Advances in Group Processes (Vol. 26, pp. 165–198). Emerald Group Publishing Limited. https://doi.org/10.1108/S0882-6145(2009)0000026010

Lo Iacono, S. (2019). Law-breaking, fairness, and generalized trust: The mediating role of trust in institutions. PLoS ONE, 14(8), e0220160. https://doi.org/10.1371/journal.pone.0220160

MacKinnon, D. P., Warsi, G., & Dwyer, J. H. (1995). A simulation study of mediated effect measures. Multivariate Behavioral Research, 30(1), 41–62. https://doi.org/10.1207/s15327906mbr3001_3

Mayer, R. C., Davis, J. H., & Schoorman, F. D. (1995). An integrative model of organizational trust. Academy of Management Review, 20(3), 709–734. https://doi.org/10.5465/amr.1995.9508080335

Mayseless, O., & Popper, M. (2007). Reliance on leaders and social institutions: An attachment perspective. Attachment and Human Development, 9(1), 73–93. https://doi.org/10.1080/14616730601151466

Moin, S. M. A., Devlin, J., & McKechnie, S. (2015). Trust in financial services: Impact of institutional trust and dispositional trust on trusting belief. Journal of Financial Services Marketing, 20(2), 91–106. https://doi.org/10.1057/fsm.2015.6

Nannestad, P., Svendsen, G. T., Dinesen, P. T., & Sønderskov, K. M. (2014). Do institutions or culture determine the level of social trust? The natural experiment of migration from non-western to Western countries. Journal of Ethnic and Migration Studies, 40(4), 544–565. https://doi.org/10.1080/1369183X.2013.830499

Robbins, B. G. (2011). Neither government nor community alone: A test of state-centered models of generalized trust. Rationality and Society, 23(3), 304–346. https://doi.org/10.1177/1043463111404665

Romano, A., Balliet, D., Yamagishi, T., & Liu, J. H. (2017). Parochial trust and cooperation across 17 societies. Proceedings of the National Academy of Sciences of the United States of America, 114(48), 12702–12707. https://doi.org/10.1073/pnas.1712921114

Rothstein, B., & Eek, D. (2009). Political corruption and social trust. Rationality and Society, 21(1), 81–112. https://doi.org/10.1177/1043463108099349

Rothstein, B., & Stolle, D. (2008). The state and social capital: An institutional theory of generalized trust. Comparative Politics, 40(4), 441–459. https://doi.org/10.2307/20434095

Rousseau, D. M., Sitkin, S. B., Burt, R. S., & Camerer, C. (1998). Not so different after all: A cross-discipline view of trust. Academy of Management Review, 23(3), 393–404. https://doi.org/10.5465/AMR.1998.926617

Seifert, N. (2018). Yet another case of Nordic exceptionalism? Extending existing evidence for a causal relationship between institutional and social trust to the Netherlands and Switzerland. Social Indicators Research, 136(2), 539–555. https://doi.org/10.1007/s11205-017-1564-x

Shrout, P. E., & Bolger, N. (2002). Mediation in experimental and nonexperimental studies: New procedures and recommendations. Psychological Methods, 7(4), 422–445. https://doi.org/10.1037/1082-989X.7.4.422

Yamagishi, T., & Mifune, N. (2008). Does shared group membership promote altruism? Rationality and Society, 20(1), 5–30. https://doi.org/10.1177/1043463107085442

Zhao, X., Lynch, J. G., & Chen, Q. (2010). Reconsidering Baron and Kenny: Myths and Truths about Mediation Analysis. Journal of Consumer Research, 37(2), 197–206. https://doi.org/10.1086/651257

Zmerli, S., & Newton, K. (2008). Social trust and attitudes toward democracy. Public Opinion Quarterly, 72(4), 706–724. https://doi.org/doi:10.1093/poq/nfn054

---

## [Decision Letter · Decision Letter 1]

23 Jun 2020

PONE-D-20-07477R1

Enhancing feelings of security: How institutional trust promotes interpersonal trust

PLOS ONE

Dear Dr. Spadaro,

Thank you for submitting your manuscript to PLOS ONE. After careful consideration, we feel that it has merit but does not fully meet PLOS ONE’s publication criteria as it currently stands. Therefore, we invite you to submit a revised version of the manuscript that addresses the points raised during the review process.

We look forward to receiving your revised manuscript.

Kind regards,

Valerio Capraro

Academic Editor

PLOS ONE

Additional Editor Comments (if provided):

I have now collected three reviews from the same reviewers who reviewed the first version of this manuscript. One review is positive and accepts the manuscript, another review is also positive and suggests a very minor revision, the third review is still negative, but leaves to the editor the ultimate decision. After reading the manuscript and the reviews, I have decided to follow the majority of the reviewers and conditionally accept this manuscript. Please address the remaining comments at your earliest convenience. I am looking forward for the final version.

Reviewers' comments:

Reviewer's Responses to Questions

**Comments to the Author**

1. If the authors have adequately addressed your comments raised in a previous round of review and you feel that this manuscript is now acceptable for publication, you may indicate that here to bypass the “Comments to the Author” section, enter your conflict of interest statement in the “Confidential to Editor” section, and submit your "Accept" recommendation.

Reviewer #1: All comments have been addressed

Reviewer #2: All comments have been addressed

Reviewer #3: All comments have been addressed

2. Is the manuscript technically sound, and do the data support the conclusions?

Reviewer #1: Yes

Reviewer #2: Yes

Reviewer #3: Yes

3. Has the statistical analysis been performed appropriately and rigorously? 

Reviewer #1: Yes

Reviewer #2: Yes

Reviewer #3: Yes

4. Have the authors made all data underlying the findings in their manuscript fully available?

Reviewer #1: Yes

Reviewer #2: Yes

Reviewer #3: Yes

5. Is the manuscript presented in an intelligible fashion and written in standard English?

Reviewer #1: Yes

Reviewer #2: Yes

Reviewer #3: Yes

6. Review Comments to the Author

Reviewer #1: Thank you for the opportunity to review this revision. The authors certainly took seriously my concerns and I appreciate their detailed responses. Unfortunately, however, I remain largely unpersuaded by the contribution of this work. It provides arguably interesting data with less than clear findings to a well-researched area in a way that just seems to muddy the water. I must leave to the editor whether this contribution is sufficient.

On the positive side, I buy the authors' argument that the idea that security might explain the institutional/interpersonal relationship is novel but it is not argued particularly well here and as a result, it is hard to see how the paper could stand on its conceptual contribution (the paper does more to say "it could be argued" than to show a concrete logic upon which future, more precise tests can rest).

Additionally, Study 2 appears pretty solid. I am curious about the operationalization of security but I understand that this is secondary data and the argument that the four institutions have some nexus with crime control is persuasive.

On the negative side, Study 1 still has huge common method variance issues (all three of the focal measures address really similar concepts--the trustworthiness of institutions, whether those institutions create feelings of security, and trust in "most Italians" [which presumably include the people who work for the institutions]). I recognize the attempt to control out these issues and the argument that this is only one piece of a larger puzzle but there still is not much "there" there.

Study 3 still has the same issues I brought up in my first review and I was largely unpersuaded by the authors' response. There is still no good reason why a participant would use information about the police in a country where an interaction partner was located to set trust in that individual UNLESS that was literally the only information to go on (as is the assumption here given that this is a manipulation). Thus, this does not represent the actual process within a country and so whatever relation is identified (even if consistent with the one in reality) happened for a different reason. Problematically though, that this manipulation still does not result in a direct effect on interpersonal trust remains a problem. I recognize and appreciate the authors' citations of work arguing that this is not a prerequisite of a significant indirect effect so I will stipulate that the indirect effect may exist but the lack of a direct effect supports my contention that the process here is different. We know that institutional trust should covary with interpersonal. That they do not suggests that something is off here.

For smaller issues, the paper still uses Mayer's definition of trust to motivate work on what those authors would call trustworthiness. This just feels unnecessary. As the authors note in the response to reviewers, they root their work in trust defined differently--why not present that as the core definition?

Additionally, the paper would benefit from a read-through for missing words, typos, and word choice.

Reviewer #2: The authors have addressed very well all points raised, and I believe that the manuscript should be published.

I have only few minor remarks:

- Table 4 (p.26) can still be improved and better show the results for the serial mediation model e.g. results for mediator 2 are not clear (in my view).

- The "General Discussion" is sometimes at odds with the "Limitations and Future Research" section. I would invite the authors to double-check for inconsistencies (even if small).

- The "Concluding Remarks" could be toned down a bit and be reviewed to be more in line with the rest of the manuscript (e.g. this sentence does not fit well with your review of the literature "Decades of research have focused on several processes that may promote trust among strangers, but very little attention has been devoted on one recurrent feature that characterize modern human interactions: the presence of institutions)".

Reviewer #3: (No Response)

7. PLOS authors have the option to publish the peer review history of their article (what does this mean?). If published, this will include your full peer review and any attached files.

Reviewer #1: No

Reviewer #2: No

Reviewer #3: No

---

## [Author Response · Author response to Decision Letter 1]

2 Aug 2020

Dear Dr. Valerio Capraro, 

Thank you for conditionally accepting our revised manuscript (PONE-D-20-07477), entitled “Enhancing feelings of security: How institutional trust promotes interpersonal trust”, to be considered for publication in Plos One. We are grateful for the helpful feedback that the reviewers have provided, and for the opportunity to improve our work.

We carefully addressed the remaining issues in this further round of revisions; we hope that this revised version can solve all the remaining concerns. In particular, in the response letter and in the manuscript, we further clarified some concern raised by Reviewer 1, and addressed the remaining issues pointed out by Reviewer 2.

For an easier navigation of the revisions, when changes have been made to the manuscript, we insert a section of text from the paper and/or provide the page numbers of the tracked version of the revised manuscript where the revisions can be found. As recommended in the submission guideline, large tables in the manuscript can be fully accessed through the “Web Layout” mode in Microsoft Word.

Editor

E: I have now collected three reviews from the same reviewers who reviewed the first version of this manuscript. One review is positive and accepts the manuscript, another review is also positive and suggests a very minor revision, the third review is still negative, but leaves to the editor the ultimate decision. After reading the manuscript and the reviews, I have decided to follow the majority of the reviewers and conditionally accept this manuscript. Please address the remaining comments at your earliest convenience. I am looking forward for the final version.

A: Thanks for conditionally accepting our manuscript. We greatly appreciate the opportunity to further improve the manuscript. We hope that you consider our revisions as raising the manuscript to the level expected at Plos One.

Reviewer #1

R1: Thank you for the opportunity to review this revision. The authors certainly took seriously my concerns and I appreciate their detailed responses. Unfortunately, however, I remain largely unpersuaded by the contribution of this work. It provides arguably interesting data with less than clear findings to a well-researched area in a way that just seems to muddy the water. I must leave to the editor whether this contribution is sufficient.

On the positive side, I buy the authors' argument that the idea that security might explain the institutional/interpersonal relationship is novel but it is not argued particularly well here and as a result, it is hard to see how the paper could stand on its conceptual contribution (the paper does more to say "it could be argued" than to show a concrete logic upon which future, more precise tests can rest).

Additionally, Study 2 appears pretty solid. I am curious about the operationalization of security but I understand that this is secondary data and the argument that the four institutions have some nexus with crime control is persuasive.

A: Thank you for your comments on the revised version of the manuscript. We are glad our revision improved your confidence on our findings regarding the contribution of this work and, in particular, the test of the mediating role of feelings of security in the relation between institutional trust and interpersonal trust (especially in light of the evidence provided in Study 2).

R1: On the negative side, Study 1 still has huge common method variance issues (all three of the focal measures address really similar concepts--the trustworthiness of institutions, whether those institutions create feelings of security, and trust in "most Italians" [which presumably include the people who work for the institutions]). I recognize the attempt to control out these issues and the argument that this is only one piece of a larger puzzle but there still is not much "there" there.

A: We understand your concern. However, based on the data and on past research, we are confident that each measure underlies a different construct, and therefore our results are not due to common method variance, for three main reasons:

(a) The correlations among those variables are not as strong to suggest that there is a concerning overlapping between constructs (ranging from r = .28 to r = .50; S2 Table, p. 12). Given the correlation between institutional trust and feelings of security (r = .50), we additionally run two confirmatory factor analyses to compare the model fit of either one or two latent factors, to rule out a possible overlap between the two variables and found evidence for a better fit of the two-factors model (see S2 Text);

(b) In Study 2 and 3 we used different operationalizations of the constructs and found a consistent pattern of results. For example, Study 2 operationalized feelings of security through a measure that is not tied to institutions, and interpersonal trust with a question that refers to “most people”; Study 3 manipulated (rather than measured) institutional trust and included a measure of trusting behavior (see S1 Table for an overview);

(c) These measures are grounded in past research and were operationalized from specific theories (e.g., Yamagishi & Yamagishi, 1994).

In the current version of the manuscript, to better clarify the different operationalizations and past research from which we selected these measures, we direct the reader to S1 Table (p. 8).

R1: Study 3 still has the same issues I brought up in my first review and I was largely unpersuaded by the authors' response. There is still no good reason why a participant would use information about the police in a country where an interaction partner was located to set trust in that individual UNLESS that was literally the only information to go on (as is the assumption here given that this is a manipulation). Thus, this does not represent the actual process within a country and so whatever relation is identified (even if consistent with the one in reality) happened for a different reason.

A: We agree with you that there is “still no good reason why a participant would use information about the police in a country where an interaction partner was located to set trust in that individual UNLESS that was literally the only information to go on”. 

That said, this is exactly what we did with our experimental manipulation when we decided not to provide the name of the partner’s country, but now realized this was not explicitly stated in the text. In the current version of the manuscript, we now add two sentences where we explicitly mention this aspect (see p. 21, 22). Hence, participants should base their trust assessments and made their trusting decision based on the information about institutions, as also done in previous research on trust and institutions (see Rothstein & Eek, 2009). 

Nevertheless, we agree that the ecological validity of the manipulation might not be optimal, but the choice of the experimental method should be understood as a complement to the other methods and data we gathered from different sources.

R1: Problematically though, that this manipulation still does not result in a direct effect on interpersonal trust remains a problem. I recognize and appreciate the authors' citations of work arguing that this is not a prerequisite of a significant indirect effect so I will stipulate that the indirect effect may exist but the lack of a direct effect supports my contention that the process here is different. We know that institutional trust should covary with interpersonal. That they do not suggests that something is off here.

A: We are glad you agree with us that the presence of the indirect effect supports one major goal of the paper. 

We agree that the lack of a null effect leaves open questions on what could have driven such null effect. However, contrary to what you suggest, the null effect itself is not enough to support specific alternative explanations (as we cannot claim that the null hypothesis is true). As we discuss in the General Discussion section, we believe that there are two potential possibilities which explain these findings. First, in our manipulation of institutional trust in Study 3, the institution does not have a direct intervening role in the game (e.g., the police do not have the power to regulate the social exchange). Another possibility is that the information about the institution learned in this context in light of a one-shot interaction is not comparable to trust assessments toward existing institutions, that are built through recurrent exposure. Both aspects are mentioned in the Limitation and future research section and the discussion of findings of Study 3 (p. 32 and p. 27, respectively).

Additionally, we now explicitly mention the lack of a main effect in Study 3 in the overall summary of the findings provided in the General discussion of the revised version of the manuscript (p. 28).

R1: For smaller issues, the paper still uses Mayer's definition of trust to motivate work on what those authors would call trustworthiness. This just feels unnecessary. As the authors note in the response to reviewers, they root their work in trust defined differently--why not present that as the core definition?

A: In the manuscript, we refer to the definition provided by Mayer and colleagues (1995) to define interpersonal trust (p. 2), as it is defined as “[...] the willingness of a party to be vulnerable to the actions of another party based on the expectation that the other will perform a particular action important to the trustor, irrespective of the ability to monitor or control that other party” (p. 712), and captures well the component related to risk and vulnerability enclosed in trust toward strangers, which is the outcome variable of all the studies presented in the manuscript. Additionally, this work motivated our choice of controlling for trust propensity as individual disposition in Study 1 and 3, as this variable is explicitly acknowledged in the model proposed by the authors. For these reasons, we would prefer to keep this citation unless the Editor has a strong preference about it.

That said, we acknowledge that we have used a different definition for institutional trust, based on the work of Devos and colleagues (2002), which is currently presented at p. 5, while discussing research on institutional trust.

R1: Additionally, the paper would benefit from a read-through for missing words, typos, and word choice.

A: Thanks for the suggestion. We made several edits across the entire body of manuscript to improve its readability and address any typo or missing word(s).

Reviewer #2

R2: The authors have addressed very well all points raised, and I believe that the manuscript should be published.

I have only few minor remarks:

- Table 4 (p.26) can still be improved and better show the results for the serial mediation model e.g. results for mediator 2 are not clear (in my view).

A: We are glad you found our revision of the manuscript adequate to address most of your concerns. 

In this revised version, we now clarify how the estimates reported in the table for mediator 1 and 2 were obtained by including a specific remark in the note:

“*Estimates of regressions of the mediators (feelings of security and trusting beliefs, respectively) predicting trusting behavior.” (p. 26)

We maintained the same structure of the table to be consistent with all the other mediation tables reported in the manuscript, but added an explicit reference to the “serial mediation models” in the heading to differentiate it from the tables summarizing models with a single mediator.

We made the same set of edits in Table B in the SI (p. 10), which summarizes a serial mediation as well.

R2: The "General Discussion" is sometimes at odds with the "Limitations and Future Research" section. I would invite the authors to double-check for inconsistencies (even if small).The "Concluding Remarks" could be toned down a bit and be reviewed to be more in line with the rest of the manuscript (e.g. this sentence does not fit well with your review of the literature "Decades of research have focused on several processes that may promote trust among strangers, but very little attention has been devoted on one recurrent feature that characterize modern human interactions: the presence of institutions)".

A: Following your comment, we revised the three sections of the General discussion section and double-checked the text to remove any inconsistencies among them and the literature review presented in the introduction. 

For example, we replaced “(...) the lack of a main effect of institutional trust in Study 3, which does not replicate what we found in Study 1 and 2” with “(...) the lack of a main effect of institutional trust in Study 3, which was observed in Study 1 and 2” (Limitations and Future Research, p. 31), as it might have seemed at odds with claims such as “That being said, the present findings appear robust and generalizable across research methodologies and variables operationalizations, and remain consistent even when controlling for relevant individual characteristics and institutional performance indicators” (Limitations and Future Research, p. 32).

We decided to completely remove the sentence you mentioned from the “Concluding Remarks” as it looked, indeed, at odds with the content in the abstract and in the literature review. Instead, we wrote a summary sentence coherent with the goals and theoretical rationale as opening of the General Discussion section (p. 27).

References:

Devos, T., Spini, D., & Schwartz, S. H. (2002). Conflicts among human values and trust in institutions. British Journal of Social Psychology, 41(4), 481–494. https://doi.org/10.1348/014466602321149849

Mayer, R. C., Davis, J. H., & Schoorman, F. D. (1995). An integrative model of organizational trust. Academy of Management Review, 20(3), 709–734. https://doi.org/10.5465/amr.1995.9508080335

Rothstein, B., & Eek, D. (2009). Political corruption and social trust. Rationality and Society, 21(1), 81–112. https://doi.org/10.1177/1043463108099349

Yamagishi, T., & Yamagishi, M. (1994). Trust and commitment in the United States and Japan. Motivation and Emotion, 18(2), 129–166. https://doi.org/10.1007/BF02249397

---

## [Editor Report · Decision Letter 2]

6 Aug 2020

Enhancing feelings of security: How institutional trust promotes interpersonal trust

PONE-D-20-07477R2

Dear Dr. Spadaro,

We’re pleased to inform you that your manuscript has been judged scientifically suitable for publication and will be formally accepted for publication once it meets all outstanding technical requirements.

Kind regards,

Valerio Capraro

Academic Editor

PLOS ONE
---

## [Editor Report · Acceptance letter]

18 Aug 2020

PONE-D-20-07477R2 

Enhancing feelings of security: How institutional trust promotes interpersonal trust 

Dear Dr. Spadaro:

I'm pleased to inform you that your manuscript has been deemed suitable for publication in PLOS ONE. Congratulations! Your manuscript is now with our production department. 

Kind regards, 

on behalf of

Dr. Valerio Capraro 

Academic Editor

PLOS ONE